



# Impact of Convectively Coupled Tropical Waves on the composition and vertical structure of the atmosphere above Cabo Verde in September 2021 during the CADDIWA campaign

Tanguy Jonville[1,2], Maurus Borne[3], Cyrille Flamant[1], Juan Cuesta[4], Olivier Bock[5,6], Pierre Bosser[7], Christophe Lavaysse[8], Andreas Fink[3], and Peter Knippertz[3]

[1]Laboratoire Atmosphères, Milieux, Observations Spatiales (LATMOS), UMR 8190, CNRS, Sorbonne Université and Université Paris Saclay, Paris, France
[2]Ecole des Ponts, 77420 Marne la Vallée, France
[3]Institute of Meteorology and Climate Research, Karlsruhe Institute of Technology, Karlsruhe, Germany
[4]Université Paris Est Creteil and Université Paris Cité, CNRS, LISA, F-94010 Créteil, France
[5]Institut de physique du globe de Paris (IPGP), UMR 7154, Université de Paris, CNRS, IGN, Paris, France
[6]Ecole Nationale des Sciences Géographiques-Géomatique, IGN, Marne-la-Vallée, France
[7]Laboratoire des sciences et technologies de l'information, de la communication et de la connaissance (Lab-STICC), UMR 6285, CNRS, ENSTA-Bretagne, Brest, France
[8]Institut des Géosciences de l'Environnement, CNRS-UGA-INRAE-IRD-Grenoble INP,38000 Grenoble, France

**Correspondence:** Tanguy Jonville (tanguy.jonville@latmos.ipsl.fr)

**Abstract.** In summer, Mixed Rossby Gravity Waves/Tropical Disturbances (MRG-TD) are known to drive intraseasonal humidity variability in the northeastern Atlantic troposphere, modulated by Equatorial Rossby (ER) and Kelvin waves. However, their impact on dust remains poorly understood, and MRG-TD tracks are often mingled in the literature. During the Clouds-Atmosphere Dynamics-Dust Interaction in West Africa (CADDIWA) campaign in September 2021, in-situ and remote sensing data (dropsondes, radiosondes, GNSS, and IASI) were used to investigate the 3D impact of tropical waves on dust and thermo-

dynamics over Cape Verde. The distinct contributions of Kelvin waves, ER, and MRG-TD were isolated using frequency-wave number filtering of Total Column Water Vapor (TCWV). The latter was efficiently split into southern and northern-track African Easterly Waves using distinct frequency windows (respectively MRG-TD1 and MRG-TD2) and enabled us to demonstrate their distinct horizontal structures and impacts. ER waves mainly impacted thermodynamics above 750 hPa, MRG-TD1 affected jet-level thermodynamics, and MRG-TD2 modulated moisture in the lower troposphere. MRG-TD2 was identified as

the main driver of dust events over Cape Verde in September 2021. Tropical cyclogenesis was linked to interactions among multiple tropical waves. Notably, a delay of up to 2 days was observed between Kelvin wave interactions with MRG-TD1 and cyclone formation, consistent with previous findings. These results highlight the critical role of tropical wave interactions in cyclogenesis and underscore their potential for improving forecasting.





## 1 Introduction

Tropical waves participate to determine synoptic and intraseasonal variability in the tropical atmosphere. They correspond to zonally propagating, equatorially trapped solutions of the Shallow Water (SW) equations, with each mode characterized by a range of wave-numbers and wave period for a given equivalent depth (Matsuno, 1966; Wheeler and Kiladis, 1999). These solutions exhibit specific horizontal structures (Matsuno, 1966; Wheeler et al., 2000), forming four categories of waves: Equa-

torial Rossby waves (ER), Equatorial Kelvin waves, Mixed Rossby Gravity waves (MRG), and Inertia Gravity waves (IG). In addition, two quasi-oscillatory phenomenon that are not predicted by the SW theory contribute strongly to the variability in the equatorial regions: the Madden-Julian Oscillation (MJO) and Tropical Disturbances (TD). In West Africa and over the North Atlantic, the African Easterly Jet (AEJ) system favour a specific type of Tropical Disturbances, namely, African Easterly Waves (AEW) that propagates north (AEW-N) and south (AEW-S) of the AEJ. Waves from each wave track present signifi-

cantly different structural (Pytharoulis and Thorncroft, 1999; Chen, 2006) and spectral structures (Jonville et al., 2024a). All together, these waves will be referred to as tropical waves in the following.

Tropical waves are responsible for a significant part of the variability of convection and precipitation above West Africa and the North Atlantic (Kamsu-Tamo et al., 2014; Lubis and Jacobi, 2015; Schlueter et al., 2019b), with TD-AEWs dominating the

convection variability in the boreal summer (Lubis and Jacobi, 2015). Kiladis et al. (2006) show how Mesoscale Convective Systems (MCS) south of the AEJ can trigger the genesis of an AEW-S which, in turn, carries and catalyses convection as the wave propagates westward (Kiladis et al., 2006; Mekonnen et al., 2006). The life cycle of the wave is modulated by precipitation above the continent with phases of wave growth (resp. decay) being correlated with more (resp. less) intense precipitation (Cornforth et al., 2009; Janiga and Thorncroft, 2013). AEW impact strongly the generation of Squall Lines, and do so more

and more as they reach the coast. Fink and Reiner (2003) found that 20% of Squall Lines generation are AEW forced at 15°E as opposed to 68% at 15°W. Although their genesis is linked to dry convection event above the Sahara (Pytharoulis and Thorncroft, 1999), the impact of AEW-N on convection during later stages of development is still debated. Kiladis et al. (2006) found that AEW-N were often associated with a convective system to the east of the trough axis, while Agudelo et al. (2011) found no significant coupling between AEW-N and deep convection. During the boreal spring, most of the outgoing longwave radiation

signal is explained by kelvin waves (Nguyen and Duvel, 2008; Sinclaire et al., 2015; Lubis and Jacobi, 2015). The active phase of convectively coupled Kelvin waves favors the initiation of convection above the continent, whereas their inactive phase prevents MCS propagation. In winter, the signal is dominated by ER waves (Lubis and Jacobi, 2015). ER waves present a more barotropic vertical structure (Kiladis and Wheeler, 1995) and tend to favour stratiform precipitation (Schlueter et al., 2019a).

Interactions between different tropical waves are frequent and affect precipitation and convection differently. Independently of the seasonality, ER and MJO control longer time-scale precipitation variability (between 7-20 days), explaining up to a third of the variance (Schlueter et al., 2019b). They have been found to modulate the amplitude of higher frequency wave. An active ER wave phase favours growth of TD-AEWs and increases convective activity inside such wave, especially above the





gulf of Guinea (Schlueter et al., 2019b). The active phase of the MJO reinforces the convective activity while slowing down
the propagation of convective systems (Laing et al., 2011). Kelvin waves favor the initiation of AEWs (Ventrice and Thorn-
croft, 2013). Lawton and Majumdar (2023) show how the passage of convectively coupled Kelvin wave increases the spatial
coverage of convection inside an AEW and low level convergence. AEWs also show a gain of moisture and relative vorticity
inside the circulation up to 1.5 days after the passage of the Kelvin wave. Due to the close vicinity of TD and MRG waves in
wavenumber-frequency space, various interactions have been reported. While "hybrid" waves have been observed over Africa
(Cheng et al., 2019), transition from MRG into off-equatorial TD waves has also been documented over the Pacific Ocean
(Takayabu and Nitta, 1993; Zhou and Wang, 2007). Tropical waves also interacts with extra-tropical waves. Interaction have
been documented between extratropical Rossby waves and ER (Schlueter et al., 2019a) and Kelvin waves (Lubis and Jacobi,
2015), but will remain out of the scope of this study.

Those interactions are especially important in shaping extreme events Lafore et al. (2017); Peyrillé et al. (2023). Lafore
et al. (2017) examined a severe precipitation occurrence in Ouagadougou on September 1, 2009 and showed how the combined
effects of a Kelvin wave, an AEW, and an ER wave contributed to its development. The merger of AEW-S and AEW-N has
been shown to favour tropical cyclogenesis (Hankes et al., 2015; Duvel, 1990; Jonville et al., 2024a). The interaction between
AEW and Kelvin wave can also catalyze cyclogenesis up to 2 days after the moisture peak associated with an AEW encounters
a Kelvin wave (Ventrice et al., 2012a; Ventrice and Thorncroft, 2013; Lawton and Majumdar, 2023). The active phase of the
MJO also creates favorable conditions for the development of a TD into a Tropical Cyclone (Frank and Roundy, 2006).

Dust is another important feature of the West African climate. It modifies the cloud structure (Saleeby et al., 2015) and has a
very significant impact on tropical cyclogenesis (Fan et al., 2016): the increase in cloud condensation nuclei or ice nucleating
particles may favour liberation of latent heat and increase convective activity (Fan et al., 2016), but the radiative effect in the
mid-troposphere stabilizes the atmosphere and reduces the sea surface temperature (Evan et al., 2006). The impact depends
on the distance between the dust-laden layer and the center of the convective activity (Shu and Wu, 2009). Dust has also been
documented to interact with TD-AEWs. TD-AEWs favor dust emission and transport (Jones et al., 2004; Knippertz and Todd,
2010; Grogan and Thorncroft, 2019; Cuesta et al., 2020; Nathan and Grogan, 2022), and dust outbreaks in turn modulate the
growth phase of TD-AEWs (Jones et al., 2004; Grogan et al., 2016, 2019). To our knowledge, the impact of the other tropical
waves on dust outbursts has not been studied.

In this context, the Clouds-Atmosphere Dynamics-Dust Interaction in West Africa (CADDIWA) campaign took place in
September 2021 (Flamant et al., 2024) with the aim of operating a suite of relevant observational and modeling devices to bet-
ter understand the interaction between dust, convection, tropical waves and tropical cyclogenesis. Ground-based and airborne
in-situ and remote sensing instruments were deployed in close synergy with satellite overpasses from Aeolus (Stoffelen et al.,
2020; Witschas et al., 2022; Borne et al., 2024) and the Infrared Atmospheric Sounding Interferometer (IASI) (Clerbaux et al.,





2009; Hilton et al., 2012).

In this study, convectively coupled tropical waves are identified based on the Total Column Water Vapour (TCWV) fields from the European Center for Medium Range Forecast (ECMWF) reanalysis (ERA5), using a space-time spectral analysis following the method described by Wheeler and Kiladis (1999); Kiladis et al. (2009) and the protocol described in Janiga et al. (2018). The latter is used to isolate ER, Kelvin waves as well as TD-AEWs. The impact of tropical waves on the 3D thermodynamic structure of the atmosphere and 3D distribution of dust is investigated using composites of radiosondes and
dropsondes launched during the CADDIWA campaign, ERA5 and CAMS reanalysis and IASI data. Especially, their impact on the tropical cyclogenesis observed during the campaign is highlighted.

## 2   Data and Method

### 2.1   Data

This study relies on observational data collected during the CADDIWA campaign (Flamant et al., 2024). A total of 44 drop-
sondes were used in 9 flights between the 9th of September 2021 and the 19th of September 2021 and 39 radiosondes were launched from the airport of Sal between the 7th of September and the 28th of September 2021 (not more than 4 per day). In-situ pressure, humidity and temperature measurements from both radiosondes and dropsondes were used to determine the vertical structure of the atmosphere during the campaign. Two Global Navigation Satellite System stations (Bock et al., 2021, GNSS) also provided for the vertically integrated water content (TCWV, Total Column Water Vapour), one permanent in Es-
pargos and the other temporary, installed in Sal for the duration of the campaign.

    Satellite data are used for integrated variables (water vapour content, aerosol optical depht ...) and for vertical profiles of extinction. The Infrared Atmospheric Sounding Interferometer (IASI) is an instrument on board the MetOp polar-orbiting satellites operated by EUMETSAT. IASI provides detailed information on atmospheric temperature, humidity, and chemistry. the
instrument captures the spectrum of the radiation emitted by the Earth and the atmosphere for a wide spectral range from 645 to 2760 $cm^{-1}$ with a spectral resolution of 0.5 $cm^{-1}$, allowing for precise retrieval of atmospheric profiles and trace gas concentrations. For Aerosol Optical Depth (AOD) and vertical profiles of dust extinction coefficient at 10 micrometers, AEROIASI (AEROsols IASI) data are used (Cuesta et al., 2015, 2020). It has been found to give results similar to the Cloud-Aerosol Lidar with Orthogonal Polarization (Cuesta et al., 2015, CALIOP, on board CALIPSO). AEROIASI is a novel IASI-based product
that allows the derivation of the dust mean layer height and the 3D dust distribution over land and ocean including the Atlantic. It depicts the vertical structure of the main dust layer in the atmospheric column and it is used in addition to the 10 $\mu$m dust AOD. TCWV from ground measurements and from reanalysis are compared. Brightness temperatures are retrieved from the Merged-IR dataset ((https://doi.org/10.5067/P4HZB9N27EKU), merging data from Japanese, European and American geostationary missions (METEOSAT-5/7/8/9/10, GMS-5/MTSat-1R/2/Himawari-8, and GOES-8/9/10/11/12/13/14/15/16), interpolated on a





grid with a spatio-temporal resolution of 4 km-30 minutes.

These observations are used in synergy with reanalysis from the European Center for Medium-range Weather Forecast (ECMWF). For physical and dynamical parameters, ERA5 data are used. ERA5 data are generated using ECMWF's Integrated Forecasting System (IFS), which assimilates a wide range of observational data, including satellite radiances, weather
station reports, radiosonde measurements, and marine observations. They have a spatial resolution of 0.25°, are gridded into 37 pressure levels (Hersbach et al., 2020), and for this study were acquired twice a day(at 0000UTC and 1200UTC). As for dust distributions, we relied on Copernicus Atmospheric Monitoring Service (CAMS), generated using ECMWF's Integrated Forecasting System (IFS), coupled with the chemistry module. Two points were taken per day, with a resolution of 0.75° and 25 pressure levels (Inness et al., 2019). It is important to note that, while CAMS assimilates data from a variety of sources, it does
not assimilate any data on the vertical distribution of dust. The dataset spans from January 2003 and therefore the climatology presented in this study is computed from 2003 to 2021.

## 2.2 Filtering

Using a Real Shallow Water model, Matsuno (1966) proposed dispersion relationships for each type of wave (see figure 1-a).
The equivalent depth defined as the depth of the real shallow water layer of the model, is determined empirically, and usually ranges from 8 to 90 meters for convectively coupled waves Wheeler and Kiladis (1999). In addition to these tropical waves, two quasi-periodic processes that are not predicted by the Real Shallow Water theory are usually discussed. Tropical Disturbances (TDs) usually encompass all types of dynamical features with a period ranging from 2 to 5 days and planetary wave number ranging from 6 to 20 (Lubis and Jacobi, 2015). They are of special importance in West Africa as they include AEWs, which
are a key feature of the region. The Madden-Julian Oscillation can also be extracted from the frequency-wavenumber space.

A wide range of identification techniques exist to highlight tropical wave activity (spatial projections, wavelet filters, parabolic cylinder functions filter or Fourier transform filter), among which the Fourier frequency-wavenumber filter is the most commonly used (Knippertz et al., 2022). It allows a straightforward attribution of each component of the signal to a theoretical wave. It can also identify coherent structures even in a random signal. Hence, the use of different input variables
(conducted here in section 3) is important to rule out filtering arctifacts. ERA5 TCWV fields are decomposed to identify the different wave signals using the methodology described in Wheeler and Kiladis (1999). The TCWV signal is projected in the frequency-wave number space using a Fourier Transform. Figure 1-b and c show the frequency-wave number spectrum for 2018-2022 normalised by a red noise. Based on either the dispersion relation from Matsuno (1966) or the spectral characteristics of TDs and MJO, each portion of the signal can be attributed to a specific process (see Figure 1).


As presented in the data section, there were no more than two flights a day and four radiosounding, resulting in sampling frequencies too low to study the highest frequency waves ( Tulich and Kiladis (2012) found for instance that the mean period



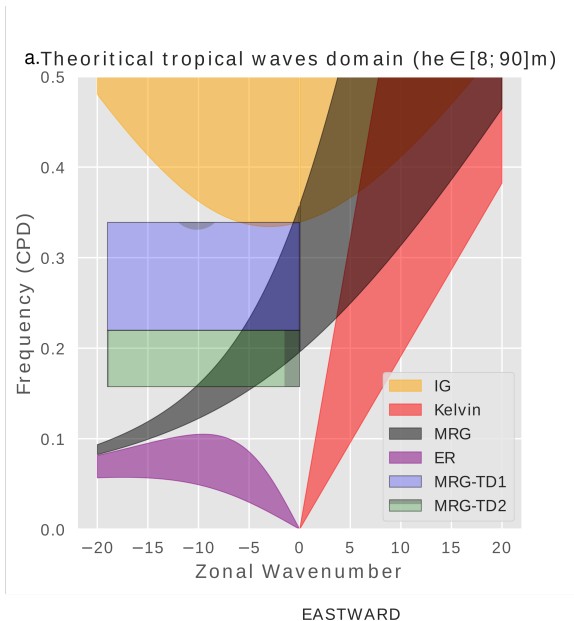

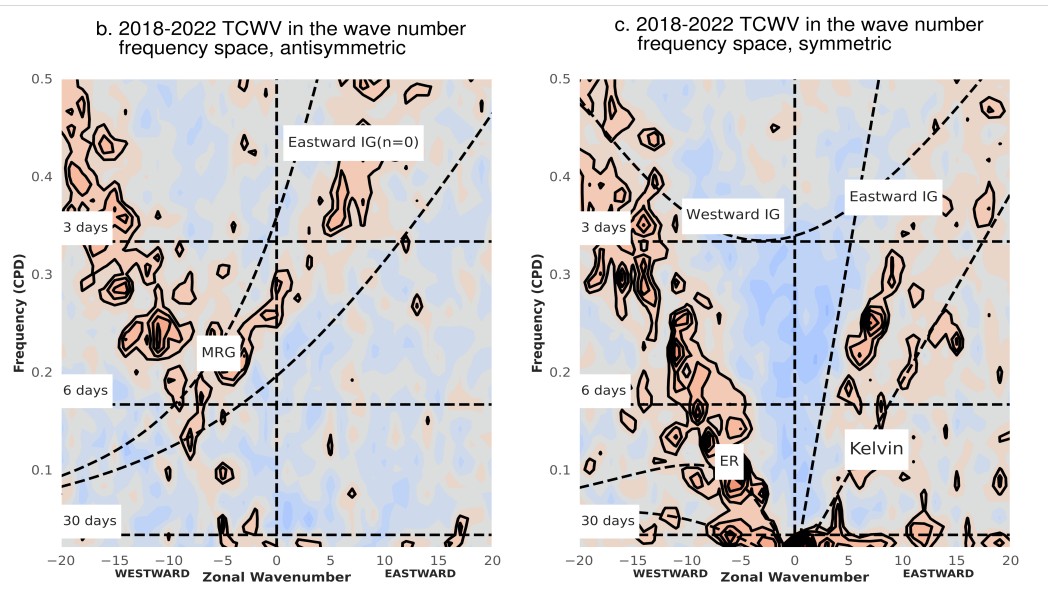

**Figure 1.** (a): theoretical domain of the different types of tropical waves according to the Real Shallow Water theory with equivalent depths ranging from 8 to 90 m. (b): 2018-2022 frequency-wave number spectrum (latitude ±30°) for antisymmetric waves normalised by a red noise. (c): same as panel b but for symmetric waves. Symmetry is understood with the equator as the axis of symmetry. This decomposition allows for a better identification of tropical waves based on their horizontal structure, as presented in Wheeler and Kiladis (1999).



of convectively coupled IG was about 8 hours). They are therefore not included in this study. For a similar reason, since the
period of the MJO ranges between 30 and 90 days (Roundy and Frank, 2004) and the campaign lasted only three weeks, the
impact of the MJO on the modulation of the dynamics, thermodynamics and composition variables was ruled out. It is however
to be noted that during these three weeks, the MJO was in an active phase over West Africa and the eastern Atlantic.

Following Jonville et al. (2024a), TDs were separated into two modes, TD1-AEW with periods between 2.95 and 4.55 days,
and TD2-AEW with periods between 4.55 and 6.35 days. Those latter modes are defined in the wavenumber-frequency space,
whereas the distinction between TD-AEW-S and TD-AEW-N is only geographical. If the present study confirms the result
of Jonville et al. (2024a), strong correlations are expected between TD1-AEW and TD-AEW-S activity and TD2-AEW and
TD-AEW-N respectively. Figure 1 shows that the domains of MRG and TDs do overlap significantly. As it has been done in
previous studies (Frank and Roundy, 2006; Janicot et al., 2010; Janiga et al., 2018), the choice is made here not to separate
MRG from TD waves. This choice was also supported by the strong similarity in the composite structure of MRG and TD2
waves (see supplementary materials for more details).

To reconstruct the contribution of the different types of wave, two-year data series of TCWV are padded with a two-year
long serie of zero, based on the padded filtering methodology of Janiga et al. (2018). The four lowest harmonics of the signal
are then suppressed to remove any inter-annual to annual trend or variability. For each wave type (ER, MRG-TD1, MRG-TD2,
Kelvin), the corresponding wave number-frequency mask is applied before an inverse Fourier transform is performed (see
fig 1-a for a visual representation of each mask).

### 2.3 Composite and significance levels

To study the impact of each wave type on the vertical thermodynamic structure of the atmosphere and the vertical dust dis-
tribution, the composites of the quantities of interest are computed based on the phase of each wave (humid or dry). For the
horizontal structure, ERA5 wind and TCWV are composited on the peaks and troughs of the 0°N-15°N averaged wave signal
at 22°W (longitude of Sal Island, Cabo Verde).

In order to study of the vertical structure, since measurements were not necessarily acquired at wave peaks or troughs, each
wave signal is decomposed into two phases: a humid phase when the wavefiltered TCWV anomaly is positive, and a dry phase
when the wavefiltered TCWV anomaly is negative. A humid (respectively dry) composite is calculated for all measurement
made during a humid (resp. dry) phase. A Welch t-test is used to assess the significance of the results (Welch, 1947). This
protocol is applied for radiosondes, dropsondes and IASI-derived vertical profiles.

The in-situ measurements offer a good coverage of the Cabo Verde region. The dropsondes were launched regularly during
campaign flights. Their positions are shown on figure 2. All the radiosondes were launched from Sal Island International
Airport. As for AEROIASI observations, the product is only available for cloud-free IASI pixels. Therefore, to prevent any



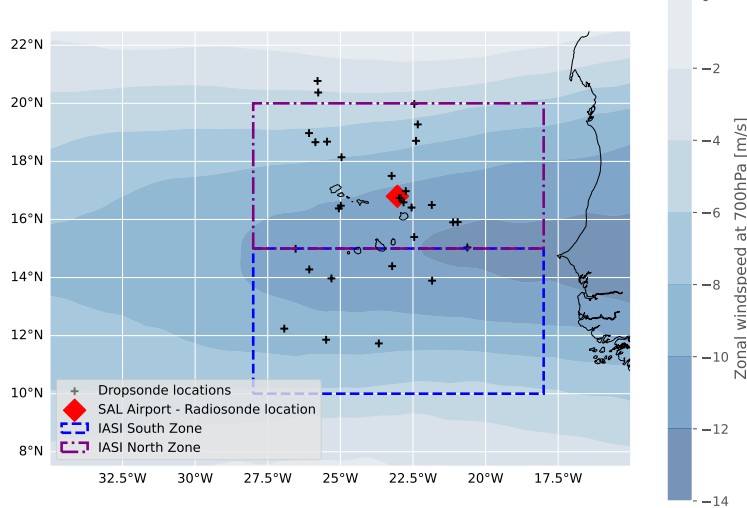

**Figure 2.** Location of radiosondes (Sal Island International Airport, red diamond), dropsondes (black crosses), IASI-CAMS composite zones (purple mixed line and blue dashed line) and mean zonal wind speed at 700 hPa in September 2021. Two composite zones are defined to study the impact of tropical waves north and south of the AEJ.

gap in the data, the information from multiple pixels need to be composited. Two zones are defined on figure 2 (IASI-CAMS South Zone and IASI-CAMS North Zone) to study the impact of tropical waves north and south of the AEJ and especially distinguish AEW-N and AEW-S structures. For a given date, wave type and phase, the only pixels that are retained are those that are cloud-free and have the signal of the given wave within the given phase (humid or dry). If these points cover less than 70% of the South or North IASI zones, the date is not used for the composite. The same method and masks are applied to CAMS data for intercomparibility reasons. A drawing is presented in section A (supplementary) for a better readability.

## 3 Horizontal structure

The horizontal structure of each wave is studied for July-August-September (JAS) 2021 using 700 hPa wind and TCWV composites at dates of peaks and troughs at 22°W of each 0°N-20°N averaged wavefiltered TCWV. This choice of latitude window is made because it is consistent with the composites obtained using the peaks at the latitude of activity of each wave (not shown). Results are presented in Figures 3 and 4.

Figures 3a and b show the composite structure of the ER wave when humid phases and dry phases are located in the vicinity of Sal, respectively. It is consistent with the structure predicted by the theory with an ITCZ located in the northern hemisphere





(Matsuno, 1966). The twin cyclonic and anticyclonic cells are well defined at about 15°N and 10°S (respectively at 30°W and 5°E on panel a, and at 30°W and 10°E on panel b). In the Northern Hemisphere, the cyclonic circulation is associated with a positive TCWV anomaly. The humid phases (resp. dry phases) are located in the rear of the circulation, where the southerly
(resp. northerly) winds predominate. At the equator, the weak positive TCWV anomalies are collocated with easterlies, whereas areas of westerlies are in deficit of TCWV. In the southern hemisphere, the impact of the ER wave on the TCWV is less defined, maybe due to the fact the the Intertropical Convergence Zone is located to the north during the boreal summer and because of our composite method, which focuses on the northern hemisphere TCWV wave signal. However a pattern can still be noted around 10°S. The ER wave is also found to strongly modulate the latitude of dust outbursts, with dusty air masses confined
above 20°N when a dry ER phase is above the ocean and dust laden air located between 10°N and 20°N when a dry phase is above Cabo Verde.

Figures 3b and c show the horizontal structure of the Kelvin wave signal. It is mostly unsignificant, very patchy and does not show coherent structures as predicted by the theory, even though the humid and dry phases show opposite patterns. The impact
on dust AOD is almost null. It is possible that our method of detection creates filtering artifact in the Kelvin wave domain, or that our region of interest is too far north (according to the theory, the dynamical center of Kelvin waves is on the equator whereas the dynamical center of ER is less constrained, see Matsuno (1966)).

Figure 4a and b show the composite structure of the MRG-TD1 wave when the humid phases and dry phases are located in
the vicinity of Sal, respectively. It is strongly consistent with the literature on AEW-S (Thorncroft, 1995; Kiladis et al., 2006; Janiga and Thorncroft, 2013; Brammer and Thorncroft, 2015). They peak at 12°N, with a northward tilt in their trajectory as they exit the West African coast. Positive anomalies of the TCWV are strongly correlated with cyclonic circulation, and negative anomalies with anticyclonic circulation. Again, the center of the TCWV anomaly lies slightly behind the dynamic center of the circulation, located either in the dominant southerly flux for the humid phase, or in the northerly flux for the dry
phase. The impact of MRG-TD1 on dust is small compared to that of ER waves or MRG-TD2.

Figures 4c and d show the composite structure of the MRG-TD2 waves. It is consistent with the dynamic highlighted by Jonville et al. (2024a) for AEW-N. The center of the dynamics is located around 21°N. For MRG-TD2 waves, positive anomalies of TCWV are located within the southerlies and negative anomalies within the northerlies. Chen (2006) documented
that the genesis of AEW-N could be linked to consecutive intrusion of the Monsoon flow (transporting moist air) and of the Harmattan (transporting dry air) across the latitude of the AEJ. This is consistent with the observed pattern of humid southerlies following dry northerlies. The dry anomaly persists as the wave crosses the Atlantic basin, although it has been documented that AEW-N become moist as they propagate over the ocean (Chen et al., 2008). MRG-TD2 northerlies are associated with strong dust outburst embedded within the dry phase of the wave. As the wave crosses the basin (not shown) it is advected by
the dynamics and can even reach humid phase areas as in panel d. This is consistent with Chaboureau et al. (2016) that found that dust emissions are predominantly associated with an Harmattan wind regime.





**Figure 3.** Composite of wave filtered TCWV (shading), 700 hPa wind (vector) for ER-related humid phases passing at 22°W during July–September 2021. CAMS wave-filtered dust AOD anomalies are shown in contours (0.04, 0.06, 0.08, 0.12 and 0.16 levels). Only values that are significant at the 10% level are shown. (a) Composite for ER humid phase at 22°W (4 events); (b) Same as panel (a) but for ER dry phases (5 events). (c) and (d) same as (a) and (b), but for Kelvin waves (24 events for each). Number of events for the composites is given in the titles.

## 4 Impact on dust and total column water vapour

As shown in the previous sections, tropical waves modulate both water vapour content and the 3D dust distribution in the Atlantic. Figures 5 and 6 show the proportion of ERA5 TCWV and CAMS dust AOD variance explained by each wave signal (squared correlation). Each figure shows the explained variance averaged between 5°W and 15°W (over the continent), and





**Figure 4.** Same as Figure 3 but for MRG-TD1 (a and b, 26 events each) and MRG-TD2 (c and d, 16 events each).

between 15°W and 25°W (over the ocean), respectively, as a function of latitude, for a September 2003-2021 climatology and for September 2021.

The climatology from September 2003 to 2021 shows that the TCVW is predominantly modulated by ER waves over West Africa and the tropical east Atlantic near the equator and around 22°N. In the region of the AEJ, MRG-TD1 and MRG-TD2 are the dominant drivers, respectively south and north of the AEJ, consistent with the dynamics of the AEWs-S and AEWs-N. Note that the peaks of explained variance for MRG-TD1 and MRG-TD2 are wider and overlap more above the Cabo Verde




region (Figure 6a,b) than above the continent (Figure 5a, b). Climatologically, the wave tracks north and south of the AEJ
are well separated above the continent, up to the longitude where the AEJ weakens (Jonville et al., 2024a). Then, mergers of
AEWs-N and AEWs-S are frequent, mainly in the eastern Atlantic (Jonville et al., 2024b; Hankes et al., 2015; Duvel, 1990)
and may explain this result. On average, Kelvin waves are the weakest contributors to the TCWV variance, explaining less than
5% of the variance accross the whole region.

Dust AOD signal is mainly explained by ER waves south of the AEJ, and by MRG-TD1 and MRG-TD2 north of the AEJ
(figure 5 and 6). This result is the same above the continent and above the eastern Atlantic. The fact that MRG-TD1 waves are
as significant a driver of AOD variance as MRG-TD2 waves is unexpected and might be due to a correlation between AEW-N
and AEW-S activity. Pytharoulis and Thorncroft (1999) documented for instance that AEW-N genesis can be triggered by
AEWs-S, creating conditions for their combined activity.

The results for September 2021 are qualitatively consistent with the climatology (Figures 5b,d and 6b, d). The signal is
noisier when only September 2021 is considered, and all explained variances are significantly higher than in the climatology.
Between 5°W and 15°W (Figure 5), south of 5°N, all wave types play a significant role in modulating the TCWV signal. South
of the AEJ, between 5°N and 13°N, the MRG-TD1 waves are the only significant drivers of the TCWV variability, explaining
25% of the variance. North of the AEJ, both MRG-TD2 and Kelvin waves signals modulate the TCWV signal with 33% and
29% of explained variance at 18°N, respectively. Further north, between 20 and 25°N, the dominant drivers are the ER waves
with 42% of the explained variance at 23°N. Above Cabo Verde (Figure 6), all wave types are also drivers of TCWV variability
between the equator and 5°N. Further north, MRG-TD1 waves are the dominant driver between 5°N and 10°N, accounting for
18% of the explained TCWV variance, while the MRG-TD2 waves dominate the signal between 10°N and 25°N with 28% of
the explained variance at 17°N. Unlike over the continent, ER waves north of 20°N and Kelvin waves at 18°N do not contribute
significantly to the TCWV variance above Cabo Verde.

For dust, the variability of the AOD signal is dominated by MRG-TD1 waves between 0 and 5°N, ER waves south of the
AEJ between 10 and 15°N, MRG-TD2 waves north of the AEJ between 15°N and 20°N and ER waves again between 20°N and
25°N above the continent. Above the Cabo Verde region, the MRG-TD2 waves strongly dominate the signal with 27% of the
explained variance between 10°N and 15°N, rising to 35% between 15 and 20°N. Weaker maxima of explained AOD variance
reaching about 15% are observed for Kelvin waves near the equator and around 13°N, for ER waves at 10°N and 23°N and for
MRG-TD2 waves at 3°N.

Irrespective of the period of time considered (climatology Vs 2021) and of the region under study (continent Vs ocean), the
main results can be summarized as follows:

- ER waves are the main driver of TCWV variance near the equator and north of 25°N, as well as the main driver of dust
  AOD variance south of the AEJ;



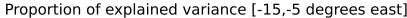

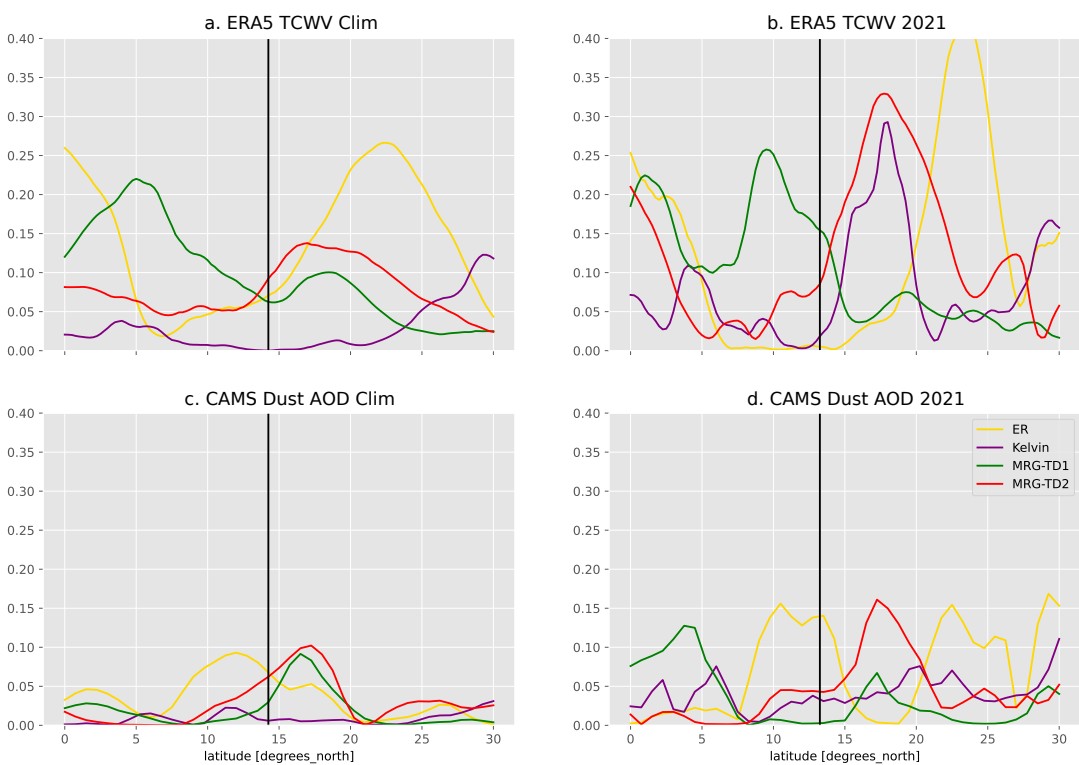

**Figure 5.** Relative importance of tropical wave signals for ERA5 Total Column Water Vapor (TCWV, panels a and b) and CAMS dust Aerosol Optical Depth (panels c and d) during September 2021 (right column, panels b and d) and September climatology for the period 2003-2021 (left column, panels a and c) above West Africa (15°W to 5°W). The explained variance is estimated by the daily squared correlations of the wave signal with the associated variable. The black line shows the mean position of the AEJ for the period and region considered.

- MRG-TD1 waves dominate in the TCWV signal variability south of the AEJ;

- MRG-TD2 waves control the TCWV signal variability north of the AEJ and also drive the dust variability north of the AEJ;

- Kelvin waves have the least significant impact on TCWV and AOD variability, except for dust north of the AEJ above the continent in September 2021.




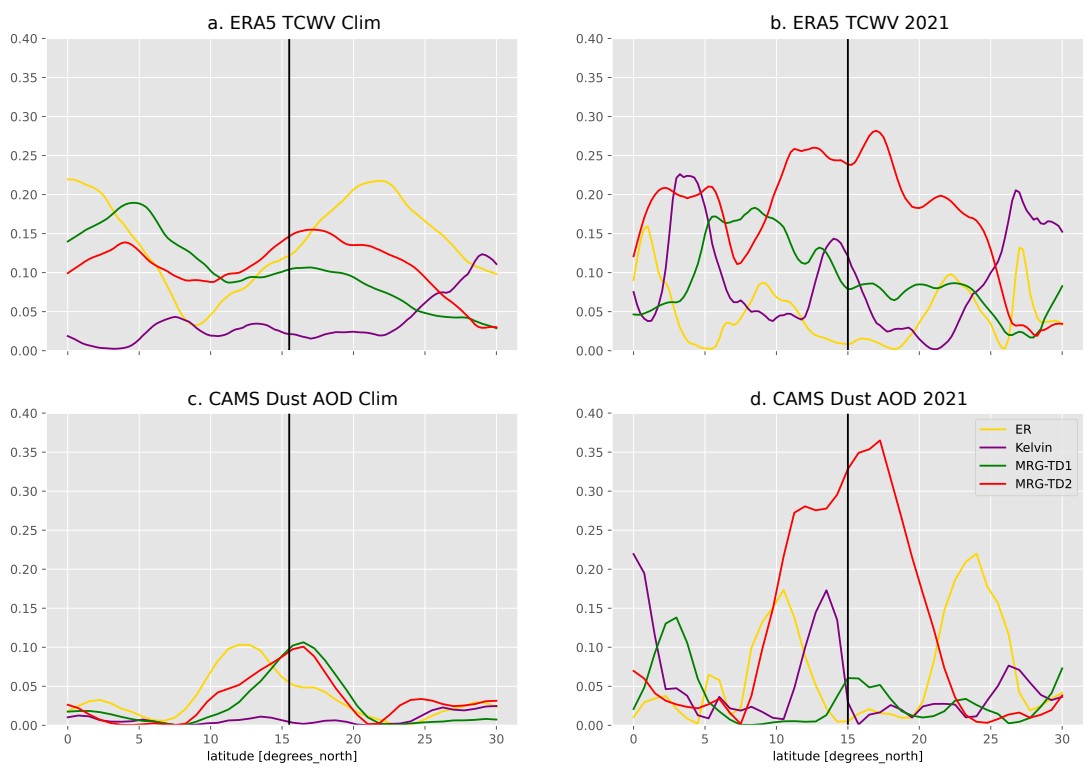

**Figure 6.** Same as Figure 5 but above Cabo Verde (between 25°W and 15°W).

## 5 Tropical waves in September 2021

Figure 7 shows the Hovmöller diagrams of ERA5-derived TCWV and brightness temperatures (from Merged-IR) in the At-
lantic basin between 5°N and 20°N with the different wave components superimposed. Deep convection ( brightness temper-
ature below 230 K) and high TCWV are almost always collocated either with a positive phase of MRG-TD1 or MRG-TD2
waves. An important dry ER event is observed during the first two weeks of September 2021 (see Figure 7a), during which
MRG-TD1 wave activity is strongly inhibited west of 20°W, but not above the continent. This is consistent with the findings
of Schlueter et al. (2019b), who show that the dry phase of an ER wave has no impact on precipitation over the continent.
No intense convection is observed during that event (Figure 7b). On the other hand, intense MRG-TD1 wave activity happens
during the humid ER phase. In particular, the genesis of the three AEW events that were the main focus of the CADDIWA
campaign were collocated with the humid ER phase (see labels PH, Peter and Rose on Figure 7). These three events were



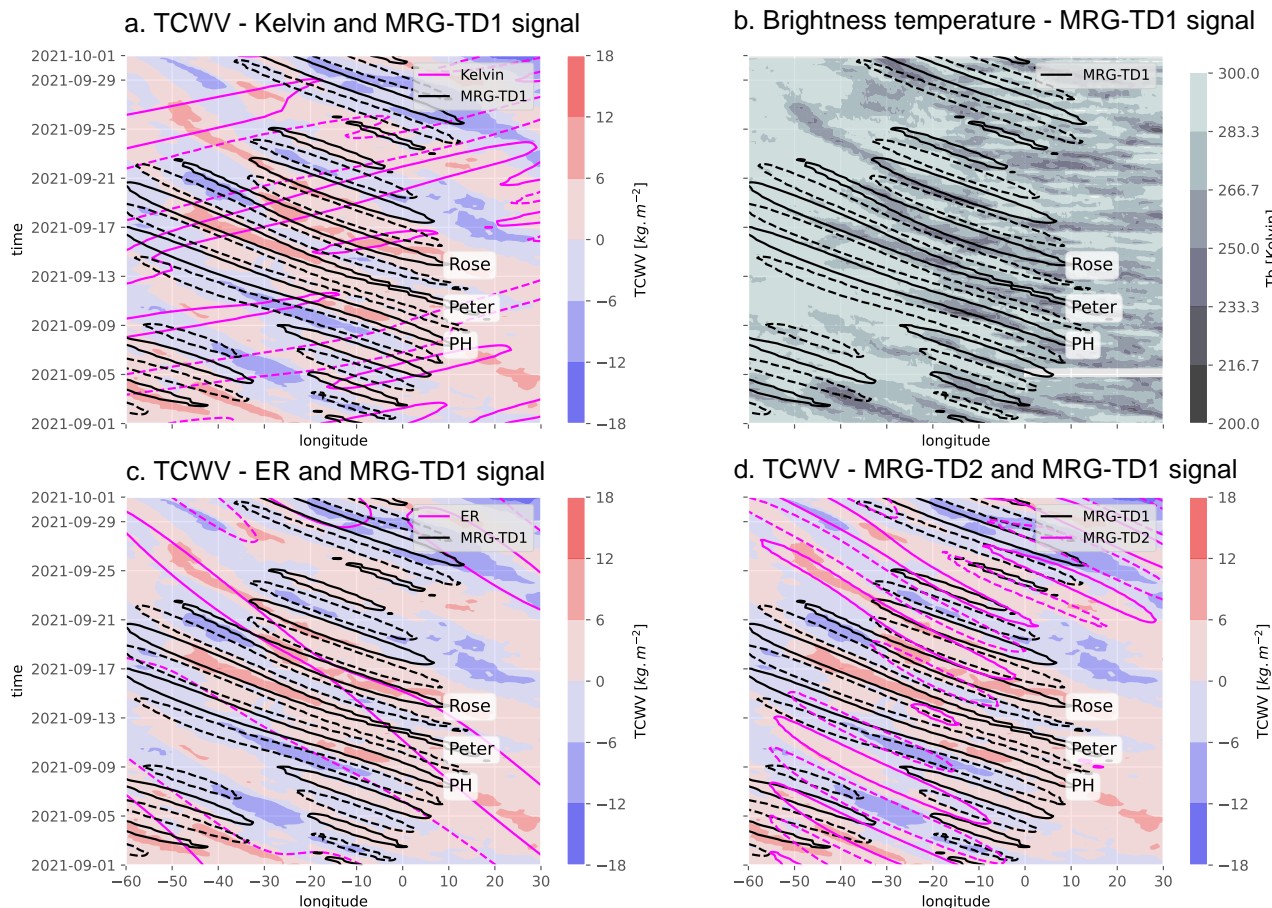

**Figure 7.** (a) Hovmöller diagram of ERA5-derived TCWV (shading) and Kelvin wave and MRG-TD1 wave signals (magenta and black contours respectively). (b) Hovmöller diagram of merged-IR brightness temperature and MRG-TD1 wave signal (black contour). (c) Same as panel (a), but for ER and MRG-TD1 waves (magenta and black contours, respectively). (d) Same as panel (a) and (c), but for MRG-TD1 and MRG-TD2 waves (magenta and black contours). Labels for MRG-TD1 associated with tropical disturbance Pierre-Henri (PH), and tropical storms Peter and Rose are shown at the start of MRG-TD1 waves propagation. Contours are drawn for plus or minus the root-mean-square value of each wave signal in September 2021. All values are averaged between 0°N and 20°N.

associated with positive TCWV anomaly and low brightness temperature, indicating strong convective activity (Figure 7b).

For tropical perturbation Pierre-Henri, the convective activity and positive TCWV anomaly fades out as the MRG-TD1 wave catches on with the dry phase of the ER wave at around 20°W, the longitude of Sal (Figure 7c). Pierre-Henri comes also close



to a MRG-TD2 dry phase, that has interacted with the MRG-TD1 wave. This case has been documented in detail from a dynamical point of view in Jonville et al. (2024b).

The disturbance from which Peter originates leaves the wettest region of the ER wave at approximately 5°W. It crosses the basin without intensification strong enough to trigger a TC genesis until it reaches 53°W (Jonville et al., 2024b). There, Jonville et al. (2024b) found that an interaction between an AEW-N and the AEW-S led to the development of a vertically coherent vortex that favoured the development of Tropical Storm Peter. The signal from this AEW-N does not appear in the MRG-TD2 signal (Figure 7d), which is consistent with the idea of Jonville et al. (2024b) that the AEW-N moistened before

merging with the AEW-S. This moistening may weaken the dry anomaly associated with the cyclonic phase and hinder the detection of this AEW-N in the TCWV signal. The MRG-TD1 wave associated with disturbance soon to become Peter also crosses a Kelvin wave humid phase on 17 September, i.e. two days before the genesis of Peter (see Figure 7a). Ventrice et al. (2012b) and Lawton and Majumdar (2023) have found that a Kelvin wave can favour TC genesis from an AEW up to two days after the two waves cross path. Peter's genesis may therefore have benefited from this interaction.


    Finally, the MRG-TD1 wave associated with the perturbation soon to become Rose benefited from a constructive interaction with the ER wave along its entire trajectory (Figure 7c) and crossed a Kelvin wave humid phase at 25°W, within a day from the TC genesis.

Figure 8a shows the TCWV time series at Sal as observed from two GNSS stations (CPVG, the permanent station of the International GNSS Service (IGS), and SAL1, the collocated temporary station installed for the CADDIWA campaign), ERA5, and radiosondes for September 2021. All three data sets are in very good agreement, especially in representing the day-to-day TCWV variability, which is quite large. A small dry bias (-0.54 $kg.m^{-2}$) is found in ERA5 compared to GNSS, with a standard deviation of 1.55 $kg.m^{-2}$, and a linear regression slope parameter close to one. The GPS and ERA5 data are available with

a 1-hour time sampling, so these statistics are rather robust. The small dry bias is also consistent with the findings of Bock et al. (2021) and of Bosser et al. (2021), for the Caribbean region in boreal autumn. The radiosonde vs. GNSS and ERA5 comparisons highlight a humid bias (1.7 to 2.0 $kg.m^{-2}$) in the radiosonde data. The number of collocated data is small in these comparisons, but the bias is significant. The inspection of observation statistics from the ECMWF operational model seems to confirm a small humid bias in the radiosonde observations in Sal. The mean observations minus analysis amounts to 0.96

$kg.m^{-2}$ (mean observations minus first-guess 1.47 $kg.m^{-2}$) (not shown).

    Figure 8b shows the AOD time series over Sal, derived from a sunphotometer part of NASA's Aerosol Network (AERONET), MODIS Terra and Aqua, as well as ECMWF Copernicus Atmospheric Monitoring Service (CAMS) reanalysis (Flemming et al., 2017). It also shows dust-only AOD (DAOD) time series from CAMS and AEROIASI. In spite of discrepancies between

CAMS, AERONET and MODIS AODs (which are partly explained by the difference in observing wavelength), these products show an overall good agreement with respect to the timing of the major dust outbreak events observed over Sal. For instance, a




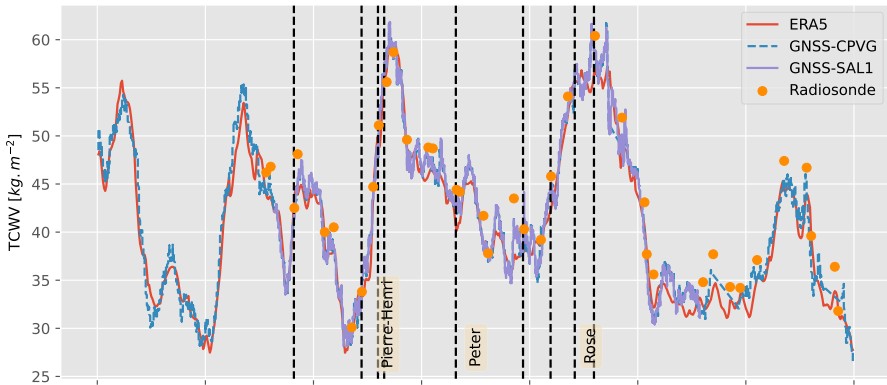

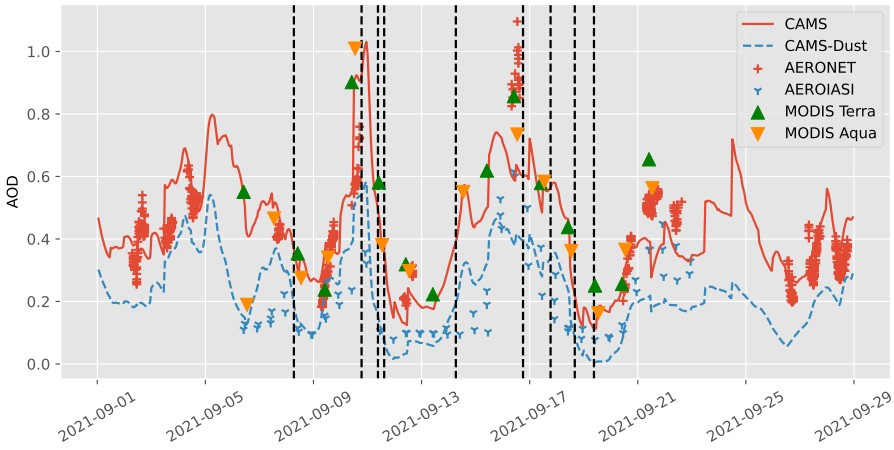

**Figure 8.** Top: TCWV time series for September 2021 at Sal as observed from two GNSS stations: CPVG (blue dashed line) the permanent station operated by INMG and IGN, and SAL1 (purple solid line), the co-located temporary station installed for the CADDIWA campaign, as well as ECMWF ERA5 (red solid line), and radiosondes launched from Sal (orange dots). Black dashed vertical lines indicate the mid-flight times of the Safire FA20 operations on 8 (flight F5), 10 (F6), 11 (F7 and F8), 14 (F9), 16 (F10), 17 (F11), 18 (F12) and 19 (F13) September 2021. The time at which the environment of TS Peter and Rose was probed in Sal is also indicated (note that Peter travelled much further south from Sal than Rose). 'Pierre Henri' refers to a tropical perturbation that passed over the Cap Verde islands but was not named or numbered by NOAA, as it did not develop into a disturbance or a storm. Bottom: Same as (a), but for AOD over Sal derived from AERONET (500 nm, red crosses), MODIS Terra and Aqua (550 nm, green and orange triangles, respectively) and CAMS 550 nm, (solid red line). Also shown are dust AOD from AEROIASI (10 $\mu$m, blue crosses) and CAMS (dashed blue line).





significantly high correlation is found, with R coefficient between AERONET AODs and AEROIASI (resp. MODIS) products is 0.95 (resp. 0.82), while the correlation between CAMS DUAOD and AERONET (resp. AEROIASI) is 0.65 (resp. 0.59).

The decomposition of the ERA5 TCWV signal is presented in Figure 9. Figure 9a shows the comparison of the sum of the wave component with the total TCWV signal. Part of the TCWV variability is not explained by tropical waves, but all the most important features can be found in the sum of tropical waves signal (correlation 61.3 %). The AOD signals and the TCWV signals are anticorrelated, with peaks in AOD coinciding with dry phases (anticorrelation between dust AOD and sum of TCWV tropical wave component -30.8%). The decomposition of TCWV into MRG-TD1, MRG-TD2, ER and Kelvin waves

is presented on panel b.

During this period, four water vapor bursts exceeding $50 \text{ kg m}^{-2}$ are seen on 1-2, 6, 11-12, and 18-19 September, respectively, and two additional ones exceeding $40 \text{ kg m}^{-2}$ are seen on 8-9 and 26 September (see Figure 8). The TCWV maximum observed on 1-2 September corresponds to the passage TS Larry (not yet a hurricane), less than 300 Nautic Miles (NM) south

of Cape Verde. It benefits from very favorable conditions, with MRG-TD1, MRG-TD2 and Kelvin waves in their humid phase and the wave filtered TCWV anomaly associated with the ER wave is null. The TCWV maxima on 11-12 September is associated with Tropical Perturbation Pierre Henri passing over Sal (Figure 9 a) in association with its parent MRG-TD1 wave in its humid phase. The wavy moisture pattern associated with the environment of Pierre Henri and TS Rose can be observed using ERA5 reanalysis (Figure 10). The humid phase associated with Pierre-Henri is collocated with a null Kelvin wave-filtered

TCWV anomaly, following a dry phase by one day. Ventrice et al. (2012b) and Lawton and Majumdar (2023) have shown that the impact of a kelvin wave on an AEW can be felt between the time the two wave cross path each other and 2 days afterwards. The Kelvin humid phase might therefore still impact unfavorably the environment of Pierre-Henri. Moreover, as already described with Figure 7, Pierre-Henri propagates in a unfavorable ER environment. The maxima on 18-19 September is associated with soon to become TS Rose propagating westward less 350 NM of Cape Verde (Figure 10), associated with

a very favorable environment (humid phase of ER, MRG-TD1 and MRG-TD2 waves, two days after a Kelvin wave humid phase) that may have catalysed the genesis of Rose in the vicinity of Sal. The TCWV peak on 6 September is not associated with a remarkable weather feature nor a TC, but could be linked to the passage of a strong MRG-TD2 wave (Figure 9). The maximum of TCWV on 8-9 September could be associated with the passage of a MRG-TD1 wave. The TCWV maximum occurring on 26 September is associated with a MRG-TD2 wave.


Regarding the AOD, all peaks are collocated with dry phases of MRG-TD2 waves. The four most intense peaks on 5, 10, 17 and 30 September are also collocated with dry phases of MRG-TD1 waves, while the less intense peak on September 23 is collocated with a humid phase of MRG-TD1 wave. The most intense dust outbreak took place late on 10 September, with AODs around 500-550 nm reaching 0.8-1. DOAD values reach 0.58 and 0.59 for CAMS (550 nm) and AEROIASI (converted from

$10 \mu m$ to 500 nm using the scaling factor of 1.7 discussed in Cuesta et al. (2020)). According to observations and simulations, the AOD and DAOD dropped significantly in the following 24 hours with the arrival of Tropical Perturbation Pierre Henri to





reach values in the range of 0.2-0.3. It is worth noting that the Safire FA20 performed 3 flights in this 2-day period (F6 through F8, see Figure 8b), sampling both high- and low-AOD conditions around Sal.

## 6 Impact on the vertical structure of the atmosphere and vertical dust distribution in September 2021

The vertical structure of the atmospheric conditions over Sal during the measurement period is now investigated using composites of temperature and humidity profiles measured in situ (radiosoundings and dropsondes) as well as dust extinction coefficient profiles derived from AEROIASI observations. Since the t-test reveals that the influence of the Kelvin waves on the vertical structure across all levels and composites is negligible, the impact of Kelvin waves will not be discussed.

Figure 11 shows the composite of the skew-T composites as a function of ER wave phase (humid or dry). The difference in the sounding skew-T (panel a) is statistically significant between 700 hPa and 600 hPa with the humid phase of the wave showing slightly colder dry temperatures and higher dew point (moister conditions), and above 550hPa being significantly hotter and showing lower dew-point.Schlueter et al. (2019a) discussed the fact that the ER waves are generally associated with stratiform precipitation. The presence of stratiform clouds between 600 and 700 hPa associated with the humid phase may explain why the upper part of the column is drier for the humid phase than the dry phase. However, it is important to note that only one humid phase and one dry phase were sampled during the campaign and the composite here may not be representative of all ER waves. As for the dust, the results are contrasted between CAMS and AEROIASI. The only significant difference found for AEROIASI is at low levels in the IASI North Zone, with a humid phase associated with a slightly more dusty events. The profiles are not significantly different in the IASI South Zone. On the other hand, CAMS detects a significant difference above 750 hPa in the IASI South Zone only. The discrepancies between the products and the small values of the differences prevent any conclusion on the impact of ER waves on dust during the campaign. It is to be noted that if the significant impact of the ER waves on the composite dust profiles in the IASI South Zone is present only in the reanalysis, the peak of ER explained variance south of the AEJ for dust AOD in Figure 5 and Figure 6 may reflect a deficiency of the model and not a realistic sensitivity of dust content to ER waves. This has to be investigated more systematically in future research.

MRG-TD1 waves have a strong impact on humidity (Figure 12a). The skew-T diagrams for the soundings and ERA5 show that the dew point is greater for the humid phase from 900 hPa to 400 hPa. The humid phase is also colder in the lower troposphere than the dry phases, from 900 hPa to 700 hPa. The difference between humid and dry phase in the sounding data is especially significant between 650 hPa and 550 hPa. The ERA5 composite shows more significance, which is to be expected as the reanalysis is less sensitive to turbulence and small scale processes, and is thus more likely to bear significant wave signal. Dust is significantly modulated by the passage of the waves in the IASI North Zone between 950 and 750 hPa for IASI observations and between 750 hPa and 550 hPa for CAMS, with more dust observed in the dry phase of the wave. On the other hand, in the IASI South Zone, MRG-TD1 waves does not impact significantly the dust distribution, neither for IASI observations nor



for the CAMS reanalysis.

MRG-TD2 waves have a strong influence on both humidity and dust distribution. The skew-T diagrams for the soundings and ERA5 show that the dew point is greater for the humid phase from 900 hPa to 550 hPa and above 500 hPa. It is significant at lower altitudes than the MRG-TD1: from 950 hPa to 700 hPa for the soundings and up to 600 hPa for ERA5. Dust is signifi-
cantly modulated by the passage of the wave throughout the column and for both IASI regions. Based on IASI observations, the extinction peak at 600 hPa disappears completely during a humid phase. However, CAMS does not detect a signal of similar magnitude, indicating a modulation in the amplitude of the peak rather than its complete disappearance. The extinction at 600 hPa during the passage of the dry phase is three times that of the humid phase for the IASI North Zone, and twice that of the humid phase in the IASI South Zone. It is important to note that the dry phase of MRG-TD2 is slightly behind the dynamic
center of the cyclonic pattern and is mainly associated with easterlies. The results are consistent with the findings of Grogan and Thorncroft (2019), who documented the fact that north of the AEJ, dust outburst are found just after the AEW trough in a region of southerly flux.

## 7 Conclusions

The CADDIWA campaign provided a unique opportunity to investigate the role of tropical waves in structuring the atmosphere in the Cabo Verde region, using an extensive range of measurements (radiosondes, dropsondes, satellite...). This research focuses on how tropical waves influence atmospheric conditions in this area, where such processes are particularly relevant due to their potential role in tropical cyclone formation.

After decomposing the TCWV signal over Sal based on ERA5 in its tropical wave components, the impact of each tropical wave type on the horizontal structure of atmospheric thermodynamics and composition is discussed. The ER waves displays a structure close to that predicted by theory. Kelvin waves exhibit a more patchy pattern, likely due to their maximum being centered at the equator. The separation between MRG-TD1 and MRG-TD2 waves allows for the detection of AEW-N and AEW-S activity, consistently with the findings of (Jonville et al., 2024a). The present study demonstrates that this method works for
integrated products in addition to the 700 hPa focus of Jonville et al. (2024a). An interesting feature of MRG-TD2 waves is that their dry phase is in quadrature with the 700 hPa wave-filtered vorticity. This supports the hypothesis of Chen (2006) that successive intrusions of moist monsoon flow (southerlies) and dry Harmattan (northerlies) across the latitude of the AEJ favors the genesis of AEW-N.

Tropical waves do not impact the whole region homogeneously. ER waves are the most significant driver of TCWV variability near the equator and north of 20°N, possibly due to interactions between ER waves and extra-tropical Rossby waves (Schlueter et al., 2019a). MRG-TD1 waves (resp MRG-TD2 waves) account for most of the impact on the atmospheric ther-





modynamics and composition profiles south (resp north) of the AEJ location for both September 2021 and climatologically. Composites of radiosoundings (and ERA5) show a significant and important impact of MRG-TD1 on the thermodynamic structure at jet level and of MRG-TD2 in the lower troposphere. These results are consistent with the regions of propagation of MRG-TD1 and MRG-TD2 (Jonville et al., 2024a). Kelvin waves have almost no impact on TCWV variability in the climatology and radiosounding composites for September 2021 show no significant difference between humid and dry phases. The consistency of ERA5 and soundings data shows that the impact of tropical waves on the thermodynamic vertical structure of the atmosphere is well reflected in ERA5.

Dust AOD variability is mainly explained by MRG-TD1 and MRG-TD2 waves north of the AEJ in the September climatology. In 2021, only the MRG-TD2 waves show an explained variance above 10% north of the AEJ. This is consistent with the satellite-based AEROIASI observations of the vertical structure of aerosol extinction. The composites show that the passage of the dry phases of MRG-TD2 waves is responsible for most of the peak in extinction in both the north and south of the AEJ, whereas the difference between the humid and dry phases of MRG-TD1 waves is insignificant for dust extinction. CAMS tends to overestimate the impact of MRG-TD1 and of ER waves on the dust extinction composite profiles. This may explain the important peak of explained variance for CAMS dust AOD south of the AEJ in both the climatology and for September 2021. CAMS also underestimates the altitude of Saharan dust compared to observation in all composites studied. Further research needs to be conducted to assess the sensitivity of CAMS to tropical waves and its performances in retrieving vertical dust profiles above West Africa. The AEROIASI products offer a good opportunity to compare CAMS results with satellite observations.

During the campaign, all MRG-TD1 waves that were associated with tropical cyclogenesis interacted with multiple other Tropical Waves. Those interactions resulted in conditions favorable to cyclogenesis. The passage of the disturbance soon to become tropical storm Larry over Sal is associated with a humid phase of a MRG-TD2 wave and a humid phase of a Kelvin wave. The collapse of tropical disturbance Pierre-Henri's convective activity happens as its associated MRG-TD1 wave catches up with the dry phase of an ER wave. The genesis of tropical storm Peter occurs two days after a MRG-TD1 wave crossed a Kelvin wave humid phase. These results are complementary to those of Jonville et al. (2024b), who documented a favorable interaction between an AEW-N and an AEW-S in the case of Peter and a detrimental interaction between an AEW-N and an AEW-S in the case of Pierre-Henri. In addition to the dynamic aspects explored by Jonville et al. (2024b), the impact of the MRG-TD2 wave needs to be considered, since the impact of MRG-TD2 waves on TCWV is in quadrature to their impact on vorticity. For tropical storm Rose, the genesis occurred near Sal, as the humid phases of an MRG-TD1 and an MRG-TD2 merge, one day after the passage of a Kelvin wave humid phase. The 0 to 2 days delay between a Kelvin/MRG-TD1 interaction and a TC genesis, documented by Ventrice et al. (2012b) and theorised by Lawton and Majumdar (2023) is consistent with our observations for both Rose and Peter. The MRG-TD1 wave associated with the genesis of TS Rose was also in phase with a TCWV ER humid phase for most of its life cycle. Jonville et al. (2024b) documented how the genesis benefited from a strong interaction with the monsoon trough. The impact of the tropical wave context on this type of interaction requires a more





thorough analysis, especially as Janicot et al. (2010) found that ER waves were strongly linked to the variability of the West
African monsoon system.


    This study explores how different equatorial waves—specifically ER, MRG-TD1, and MRG-TD2—shape humidity, tem-
perature, and dust distribution over Cabo Verde in September 2021. By analyzing in-situ temperature and humidity profiles,
dust measurements from satellite data, and reanalysis, this research highlights significant wave-induced variations in atmo-
spheric moisture and dust concentrations. Discrepancies in the dust distribution between satellite products and reanalysis data

underscore the need to further refine the modelisation of dust transport above the eastern Atlantic. This matter is particularly
relevant in the context TC genesis as the impact of dust on TC formation is still poorly understood. Notably, interactions of
MRG-TD1 and MRG-TD2 waves are known to favor TC development Hankes et al. (2015); Duvel (2021); **?**. Investigating
how dust transported by an MRG-TD2 interacts within an environment made favorable for convection by an MRG-TD1 could
help improve our understanding of TC genesis processes in the Atlantic.

**Appendix A: Method for IASI composites**

A schematic is presented in figure A1 to illustrate the method used to composite IASI data for vertical profiles. IASI and CAMS
data are first interpolated on the same grid. For a given date, wave type, and phase, only IASI cloud-free pixels with the signal
of the specified wave in the given phase (humid or dry) are retained. The same mask is applied to CAMS data. If these points
cover less than 70% of the zone of interest, the date is excluded from the composite.

*Author contributions.* CF planned the campaign, acquired the funding and administrated the project. CF, CL, PK, AF were responsible for
the supervision of this research. TJ, MB, JC, OB, PB, CL carried out the measurements and prepared the data. TJ and MB developed the
methodology and analized the data. CF and CL validated the methodology and results. The original draft was prepared by TJ, MB and CF.

*Code and data availability.* Data and code available on request from the authors.

*Competing interests.* The authors declare that they have no conflict of interest.

*Financial support.* Centre National d'Etudes Spatiales (CNES); European Space Agency (ESA, RFP/3-16595/20/NL/FF/ab); National pro-
gram LEFE of the Institut des Sciences de l'Univers (INSU) of Centre National de la Recherche Scientifique (CNRS); Institut Pierre-Simon
Laplace (IPSL); Météo France; Ministère de la Transition Ecologique et de la Cohésion des Territoires (MTECT).



*Acknowledgements.* Airborne data were obtained using the aircraft managed by Safire, the French facility for airborne research, an infrastructure of the Centre National de la Recherche Scientifique (CNRS), the Centre National d'Etudes Spatiales (CNES) and Météo-France. The airborne component of CADDIWA was supported by the national program LEFE of CNRS-INSU, CNES, the European Space Agency (ESA) and the Institut Pierre-Simon Laplace (IPSL). The AEROIASI products have been developed thanks to the support of CNES and the Programme National de Télédétection Spatiale (PNTS) of CNRS-INSU. We acknowledge the data centre AERIS (https://www.aeris-data.fr) for providing the level 1 data of IASI (originally supplied by EUMETSAT; http://www.eumetsat.int). Meteorological reanalyses produced by ECMWF are supplied by CLIMSERV (http://climserv.ipsl.polytechnique.fr). Tanguy Jonville acknowledges the French Ministry for the Environmental Transition (MTECT) for the funding of his PhD.



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





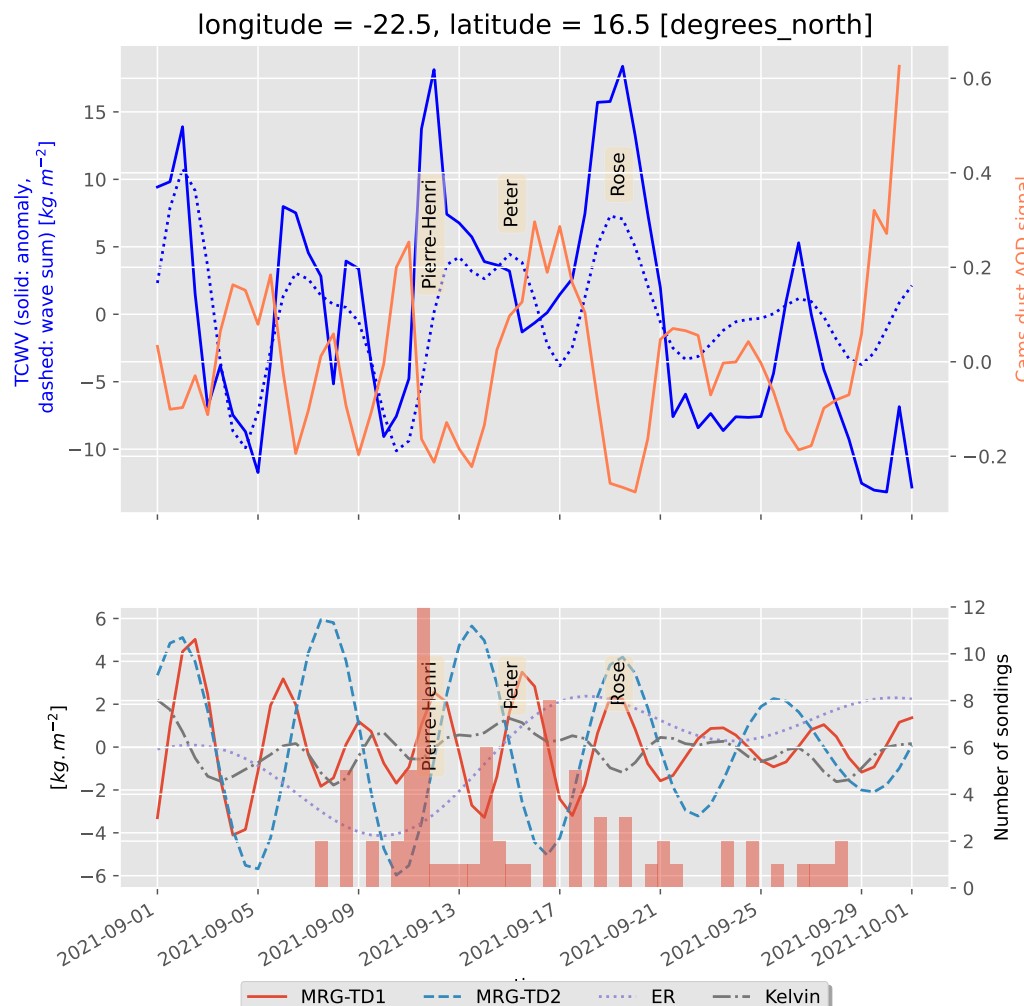

**Figure 9.** Total column water vapour (TCWV) anomaly (solid teal line, ECMWF ERA5), dust aerosol optical depth (AOD) anomaly (solid coral line, ECMWF CAMS) and sum of tropical wave contributions (dashed teal line) (a) together with individual tropical wave contributions (b) filtered following the Wheeler-Kiladis approach during September 2021 near Sal Island [16.5°N;22.5°W]. The anomalies are computed relative to the period 2003-2021. The names of the identified weather events (hurricanes 'Larry', tropical perturbation 'Pierre Henri' and TS 'Peter' and 'Rose') crossing Sal are indicated in panel a).



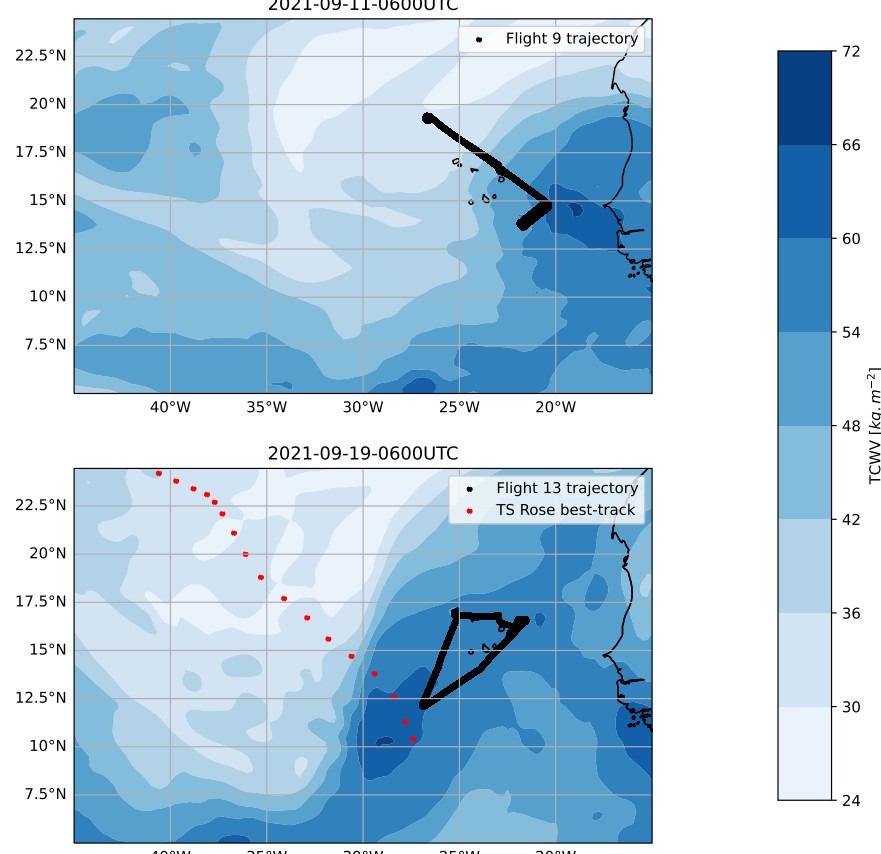

**Figure 10.** (a) Spatial distribution of TCWV from ERA5 with the Safire FA20 flight tracks (black line) conducted during the tropical waves (combination of TD/AEW and MRG) episode on 11 September in the morning. (c) Same as (b), but for 19 September 2021. The red dotted line indicates the best track estimates of TS Rose as detected by the National Hurricane Center in the domain..





**Figure 11.** (a) Skew-T composite of soundings for the humid and dry phases of ER wave. (b): Skew-T composite of ERA5 data interpolated on the same date and location as the soundings. (c) AEROIASI vertical dust extinction composite for the humid and dry phases of ER wave on IASI North Zone. (d) Same as (c) but for CAMS dust mixing ratio. (e) Same as (c) but for IASI South Zone. (f) Same as (d) but for IASI South Zone. The grey zones show where a significant gap in the two profiles is found, according to a Welch t-test (95% confidence level). Number of soundings in dry (resp. humid) phase: 43 (resp. 34). Number of days in IASI North Zone in dry (resp. humid) phase: 20 (resp 17). Number of days in IASI South Zone in dry (resp. humid) phase: 23 (resp 17).



**Figure 12.** Same as figure 11 but for MRG-TD1 waves. Number of sondings in dry (resp. humid) phase: 40 (resp. 37). Number of days in IASI North Zone in dry (resp. humid) phase: 11 (resp 10). Number of days in IASI South Zone in dry (resp. humid) phase: 16 (resp 19).







**Figure 13.** Same as figure 11 and 12 but for MRG-TD2 waves. Number of sondings in dry (resp. humid) phase: 45 (resp. 32). Number of days in IASI North Zone in dry (resp. humid) phase: 19 (resp 20). Number of days in IASI North Zone in dry (resp. humid) phase: 17 (resp 14).



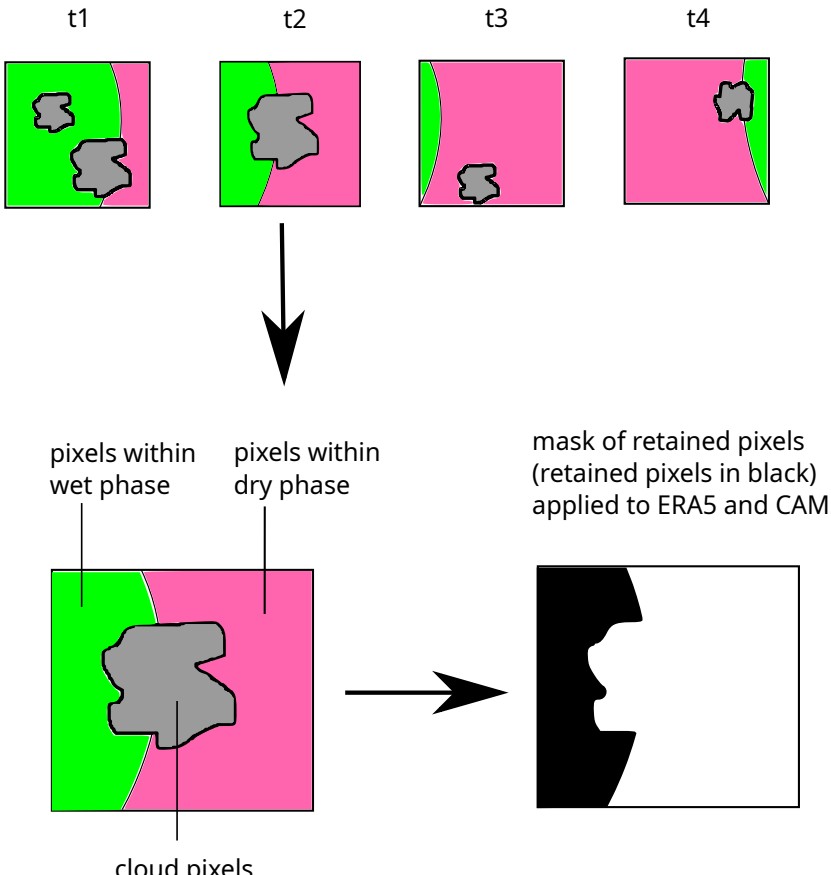

**Figure A1.** Schematic for the IASI composite method: compositing for the humid phase (in green). The phase of the wave signal is computed for each pixel. Then, for each time step, cloud pixels (in grey) and pixels in a dry phase (in pink) are masked out. All retained pixels are then averaged for both IASI and ERA data (reinterpolated on IASI grid) to create the composites.