# Peer review of "Impact of Convectively Coupled Tropical Waves on the composition, the vertical structure of the atmosphere and Tropical Cyclogenesis in the region of Cabo Verde in September 2021 during the CADDIWA campaign"

_EGUsphere, 2024_

## Referee Comment (RC1)

**Impact of Convectively Coupled TropicalWaves on the composition and vertical structure of the atmosphere above Cabo Verde in September 2021 during the CADDIWA campaign**

**Tanguy Jonville, Maurus Borne, Cyrille Flamant, Juan Cuesta, Olivier Bock, Pierre Bosser, Christophe Lavaysse, Andreas Fink, and Peter Knippertz**

The manuscript analyzes the relative impact of convectively coupled equatorial waves during September 2021 on the atmospheric state over the eastern tropical Atlantic. The analysis is thorough and presented in great detail. The current presentation is fairly dense and it goes to the detriment of clear communication of the mean takeaways. It would be helpful for the reader if the authors tried to revise the manuscript with the goal in mind to get the main points communicated very clearly. My recommendation would be minor revision, because the revisions (and possibly restructuring) I'm asking for would not involve any new analysis. Detailed comments are below.

**Comments**

1. Section 2.1 Data: This section would really benefit from a detailed table that includes all data sources, resolution, variables, time period for the climatology if applicable, etc. It's hard to follow all the details and differences in temporal availability in the text.

2. Page 7, lines 166-167: I'm worried that using a Kelvin filter that includes frequencies higher than 0.3cpd is essentially filtering noise. Figure 1c shows that even for 5 years 2018-2022 there is no coherent signal above 0.3cpd. Taking this together with Figure 3c and 3d, I wonder whether there is actually a Kelvin signal present during this time? Wheeler and Kiladis (1999) cut off the Kelvin wave filter at 0.4cpd and wave number 14 to exclude those regions that don't have a lot of Kelvin wave power. It is not clear what exactly is done here, as the mask in Figure 1a is larger than the region used in Wheeler and Kiladis (1999).

**Minor comments**

1. Page 2, line 25: "... different structural (Pytharoulis and Thorncroft, 1999; Chen, 2006) and spectral structures..." This is not clear, please rephrase. I assume the sentence is supposed to emphasize the spatial and temporal differences in the structure of the northern and southern tracking AEWs?

2. Page 2, line 29: "TD-AEWs" This acronym has not been defined yet.

3. Page 2, lines 47-48. "... of higher frequency *waves*." and "... convective activity inside such *waves*,..."

4. Page 5, line 133: "... 2 to 5 days and *westward* planetary wave number..."

5. Page 5, line 143: Do you mean a red noise spectrum?

6. Page 7, line 163: "... wave, two-year data series of TCWV are padded..." Which two years? Please specify.

7. Page 7, line 164: "... long *series* of zero..."

8. Page 7, line 174: "In order to study the vertical ..."

9. Page 9, lines 211-212: I don't think this is the case. Figure 7b here (`https://link.springer.com/article/10.1007/s00382-009-0697-2`) shows that Kelvin filtered precipitation variance peaks around 5N in the Atlantic.

10. Page 13, lines 282-283: Figure 5b shows significant Kelvin wave variance over the continent. Or maybe I am misunderstanding something?

11. Page 14, lines 287: "...September 2021 (see Figure 7a), during..." Should this be Figure 7$c$?

12. Page 15, line 295: "For tropical perturbation Pierre-Henri, ..." I assume this is the same as the AEW labeled PH? This should be mentioned here or above. I see it is mentioned in the figure caption, but I would suggest also mentioning it in the main text. I missed the caption the first time.

13. Page 22, line 475: "... modelisation..." Not sure that this is a word in english, modeling or model development would be more appropriate.

14. Page 22, line 477: "... development Hankes et al. (2015); Duvel (2021); ?." There seems to be a citation missing?

**Figures**

**Figure 1:** For panels b and c what are the contours? This needs a colorbar or labels on the contour lines.

**Figure 5:** The title should have units degrees west instead of negative degrees east. The y-axes need labels.

**Figure 11:** The caption reads: "The grey zones show where a significant gap in the two profiles is found,..." I'm having a hard time figuring out which of the two greys in the background is the significant one. Also, does this mean significant difference in the temperature or dewpoint soundings? They can't always be both significantly different at the same levels?

---

## Referee Comment (RC2)

**Review of egusphere-2024-3606**

**Title:** Impact of Convectively Coupled Tropical Waves on the composition and vertical structure of the atmosphere above Cabo Verde in September 2021 during the CADDIWA campaign

**General comment**

The manuscript discusses the impacts of convectively coupled tropical waves on the atmosphere in the eastern Atlantic regions. This study analyzed the interaction between these waves in detail. However, the current manuscript is somewhat dense and difficult to follow. Some viewpoints would benefit from further elaboration and discussion. I recommend a major revision, as revising or reorganizing the content would help readers better understand the significance of this work. My detailed comments are below.

**Specific comments**

1.  The authors should explain why these four events (Larry, Pierre Henri, Peter, and Rose) are discussed in this study and how they are defined. This could be clarified further in Section 2. In addition, 'Larry' is not labeled in any figure (it only appears in the caption of Figure 9). Furthermore, if the authors primarily focus on the time period of these events, presenting only the time range from September 1 to September 21 (or September 25) in Figures 7–9 may make the results clearer.

2.  In Figures 5 and 6, the ER waves strongly drive the AOD variance south of the AEJ (10°N–15°N), where their contribution to the TCWV variance is relatively weak. Is this because ER waves typically have strong zonal winds south of the maximum TCWV anomalies (Figures 3a and 3b)? The impact of wind anomalies on AOD variance may also be worth discussing in the main text.

3.  I got lost many times while reading Sections 5 and 6, perhaps because many figures are discussed interchangeably without clear figure references. Additionally, in Figures 11–13, I am curious whether the differences between the phase composites for MRG-TD waves (ER waves) could also be influenced by ER waves (MRG-TD waves). Since multiple wave activities coexist simultaneously (Figure 7), the phase composites of the sounding raw data would be affected by all of these waves (Figures 11–13).

**Minor comments**

1. Line 30: "Kiladis et al. (2006) *showed*…"

2. Line 40: "…by *Kelvin* waves (…"

3. Line 121: There should be a space between "..twice a day" and "(at 0000UTC and 1200UTC)"

4. Line 129: "…(see *Figure 1a*)"

5. Line 131: "…coupled waves *(Wheeler and Kiladis, 1999)*"

6. Line 135 and Figure 1: The MJO is not labeled in the Figure 1.

7. Line 167: "*Fig. 1a* for a …"

8. Line 181: "…shown on *Figure 2*…"

9. Line 250: "(*Figures* 5 and 6)…"

10. Lines 267-271: This paragraph discusses the results from Figures 5d and 6d. Thus, I would suggest labeling 'Figure 5d' and 'Figure 6d' to make the text clearer for readers. In addition, in Line 271, I believe the authors meant to refer to MRG-TD1 rather than MRG-TD2 waves (at 3°N).

11. Lines 276 and 431: I would suggest that ER waves are the main driver of TCWV variance near the equator and "*north of 20°N*", because one of their peaks is observed within 20°N-25°N (Figures 5b and 6b).

12. Line 281: According to Figure 5b, I suspect the authors meant to refer to "except for *TCWV* north of the AEJ above the continent in September 2021"

13. Line 287: It is for ER event, so it should be "(see Figure 7c)" rather than Figure 7a.

14. Figure 7 caption: I think that the shadings are TCWV "anomalies", which should be noted in caption. In Lines 3-4 of Figure 7 caption, it should be "but for MRG-TD1 and MRG-TD2 waves (*black and magenta contours*)"

15. Lines 344-345: How do we know TS Larry less than 300NM south of Cape Verde? Perhaps it needs to be marked as "not shown"

16. Line 347: "…(Figure *9a*).."

17. Line 351: "…of a *Kelvin* wave on…"

18. Line 477: "…; Duvel (2021); ?…" is it missing the reference?

19. Figures 11-13. Line labels below the panel a and b are different. (Below panel b) Dashed lines should be dew point, right?

---

## Author Comment (AC1)

**Review of the Paper « Impact of Convectively Coupled Tropical Waves on the composition and vertical structure of the atmosphere above Cabo Verde in September 2021 during the CADDIWA campaign »**

*Now : « Impact of Convectively Coupled Tropical Waves on the composition, the vertical structure of the atmosphere and Tropical Cyclogenesis in the region of Cabo Verde in September 2021 during the CADDIWA campaign »*

Dear Editor

Thank you for allowing us to enhance the paper quality. We appreciate the reviewers' valuable assistance and thoughtful feedbacks.  All the suggested modifications from the reviewers have been incorporated. The structure has been extensively revised, as suggested by the reviewers, to improve clarity. Figures have been reordered accordingly, previous Figures 11, 12 & 13 are now compacted in Figures 8 & 9 and former Figure 10 has been discarded to improve readability and make the article more straightforward.

Please find in the following the detailed answers to the reviewer's comment. Text in red indicates direct quotes from the manuscript.

| Reviewer #1 |
| --- |

*1) Section 2.1 Data: This section would really benefit from a detailed table that includes all data sources, resolution, variables, time period for the climatology if applicable, etc. It's hard to follow all the details and differences in temporal availability in the text.*

Table added in the text as required, and former figure 8 that compares all data sources has been moved to the data section and now is discussed earlier in paper.

*2) Page 7, lines 166-167: I'm worried that using a Kelvin filter that includes frequencies higher than 0.3cpd is essentially filtering noise. Figure 1c shows that even for 5 years 2018-2022 there is no coherent signal above 0.3cpd. Taking this together with Figure 3c and 3d, I wonder whether there is actually a Kelvin signal present during this time? Wheeler and Kiladis (1999) cut off the Kelvin wave filter at 0.4cpd and wave number 14 to exclude those regions that don't have a lot of Kelvin wave power. It is not clear what exactly is done here, as the mask in Figure 1a is larger than the region used in Wheeler and Kiladis (1999)*

This has been clarified in the text (l193-196). The cut offs at 0.4 cpd and wave number 14 were indeed implemented following Wheeler and Kiladis (1999).

**Specific comments**

1. Page 2, line 25: "... different structural (Pytharoulis and Thorncroft, 1999; Chen, 2006) and spectral structures..." This is not clear, please rephrase. I assume the sentence is supposed to emphasize the spatial and temporal differences in the structure of the northern and southern tracking AEWs?

This sentenced has been rephrased as:

> They will be referred to as TD-AEW in the following. Waves from each wave track present significant differences in their horizontal and vertical structure (Pytharoulis and Thorncroft, 1999; Chen, 2006), as well as different period and wavelength (Jonville et al., 2024a).

2. Page 2, line 29: "TD-AEWs" This acronym has not been defined yet.

Definition of the acronym added L24.

3. Page 2, lines 47-48. "... of higher frequency waves." and "... convective activity inside such waves,..."

Corrected in the text.

4. Page 5, line 133: "... 2 to 5 days and westward planetary wave number..."

Corrected in the text.

5. Page 5, line 143: Do you mean a red noise spectrum?

Corrected in the text.

6. Page 7, line 163: "... wave, two-year data series of TCWV are padded..." Which two years? Please specify.

Corrected in the text.

7. Page 7, line 164: "... long series of zero..."

Corrected in the text.

8. Page 7, line 174: "In order to study the vertical ..."

Corrected in the text.

9. Page 9, lines 211-212: I don't think this is the case. Figure 7b here (https://link.springer.com/article/10.1007/s00382-009-0697-2) shows that Kelvin filtered precipitation variance peaks around 5N in the Atlantic.

The reference to the article has been added in the text. The Kelvin wave composite is now computed based on the 10°S-10°N averaged TCWV decomposition to better capture the wave structure. This adjustment results in a peak around 5°, consistent with the findings of the referenced article. This is discussed L248.

10. Page 13, lines 282-283: Figure 5b shows significant Kelvin wave variance over the continent. Or maybe I am misunderstanding something?

Indeed, precision added in the text.

11. Page 14, lines 287: "...September 2021 (see Figure 7a), during..." Should this be Figure 7c?

Corrected in the text.

12. Page 15, line 295: "For tropical perturbation Pierre-Henri, ..." I assume this is the same as the AEW labeled PH? This should be mentioned here or above. I see it is mentioned in the figure caption, but I would suggest also mentioning it in the main text. I missed the caption the first time.

Indeed, PH is the acronym for Pierre-Henri. Precision on the names of the events added at the beginning of section 6.

13. Page 22, line 475: "... modelisation..." Not sure that this is a word in english, modeling or model development would be more appropriate.

Corrected in the text.

14. Page 22, line 477: "... development Hankes et al. (2015); Duvel (2021); ?." There seems to be a citation missing?

Corrected in the text.

Figure 1: For panels b and c what are the contours? This needs a colorbar or labels on the contour lines.

Colorbar added to the figure.

Figure 5: The title should have units degrees west instead of negative degrees east. The y-axes need labels.

Figure modified according to the recommendation.

Figure 11: The caption reads: "The grey zones show where a significant gap in the two profiles is found,..." I'm having a hard time figuring out which of the two greys in the background is the significant one. Also, does this mean significant difference in the temperature or dewpoint soundings? They can't always be both significantly different at the same levels?

Area are marked in dark grey on the Skew-T when either one of the two parameter is significantly different between the two phases. This allow identifying vertical levels where there is a significant difference in the thermodynamic structure, related to either temperature or humidity. Explanation added in the caption.

| Reviewer #2 |
| --- |

The manuscript discusses the impacts of convectively coupled tropical waves on the atmosphere in the eastern Atlantic regions. This study analyzed the interaction between these waves in detail. However, the current manuscript is somewhat dense and difficult to follow. Some viewpoints would benefit from further elaboration and discussion. I recommend a major revision, as revising or reorganizing the content would help readers better understand the significance of this work.

The paper has been extensively restructured. Especially, a better distinction was made in the text between the impacts of the tropical waves on the structure of the atmosphere and on tropical cyclogenesis. The mention to tropical cyclogenesis has been added in the tittle. Some paragraph were added to improve the clarity of the message and some figures have been removed or simplified.

The authors should explain why these four events (Larry, Pierre Henri, Peter, and Rose) are discussed in this study and how they are defined. This could be clarified further in Section 2. In addition, 'Larry' is not labeled in any figure (it only appears in the caption of Figure 9). Furthermore, if the authors primarily focus on the time period of these events, presenting only the time range from September 1 to September 21 (or September 25) in Figures 7–9 may make the results clearer.

All discussions relative to Tropical Cyclogenesis have been moved to section 6 and better introduced (cf L360-367) :

> The sections above have shown that Tropical Waves play a role in structuring the thermodynamics and composition of the atmosphere. In this section, the impact of Tropical Waves interplay on tropical cyclogenesis and dust outburst will be highlighted. To do so, two viewpoints will be adopted : Figure 10 present Hovmöllers each type of waves on the whole Atlantic basin, and Figure 11 shows the variation of TCWV and dust AOD at the location of Sal, Cabo Verde, in relationship with time series of wave filtered signal at the same location. Four events of interest were captured by the instruments of the CADDIWA campaign. are TC Larry, TD Pierre-Henri (PH, unofficial name used in all papers of the CADDIWA campaign, see Flamant et al. (2024); Jonville et al. (2024a)), TS Peter and TS Rose (named by the National Hurricane Center). The environments of the MRG-TD1 on which the last three event developed were sampled by the aircraft during the campaign, while Larry was only observed from ground-based measurements.

The label for hurricane Larry has been added to figure 11 (formerly figure 9). The trajectories of the convective system and TC genesis have also been added to the Hovmöller (figure 10, formerly figure 7). The choice was made to keep all events in the time window, especially to discuss the impact ER waves before and during the active phase of Pierre-Henri.

2. In Figures 5 and 6, the ER waves strongly drive the AOD variance south of the AEJ (10°N–15°N), where their contribution to the TCWV variance is relatively weak. Is this because ER waves typically have strong zonal winds south of the maximum TCWV anomalies (Figures 3a and 3b)? The impact of wind anomalies on AOD variance may also be worth discussing in the main text.

Thanks for the suggestion. A discussion was added in the main text on this specific point L240-241:

> This might be explain by the strong westerlies associated with humid phase of ER (see Fig 4a) south of the AEJ, that prevent exit of air masses from the continent.

3. I got lost many times while reading Sections 5 and 6, perhaps because many figures are discussed interchangeably without clear figure references. Additionally, in Figures 11–13, I am curious whether the differences between the phase composites for MRG-TD waves (ER waves) could also be influenced by ER waves (MRG-TD waves). Since multiple wave activities coexist simultaneously (Figure 7), the phase composites of the sounding raw data would be affected by all of these waves (Figures 11–13).

As presented above, section 5 and 6 were restructured and simplified to improve clarity. As for the cross-talk between a phase composite and another type of wave, the distribution of soundings ensures that the composites are not too biased. Figure 11 shows this distribution. Especially, for MRG-TD2, two peaks of TCWV and one trough were sampled during a negative ER phase, and two peaks and two troughs during a positive ER phase. For MRG-TD1, 2 peaks and 3 troughs

were sampled during ER negative phase, one peak is collocated with a neutral phase and 3 peaks and 3 troughs during the positive phase. There is therefore a good balance of all ER conditions in the composites of MRG-TD, and vice versa.

**Specific comments**

1. Line 30: "Kiladis et al. (2006) showed…"

Corrected in the text.

2. Line 40: "…by Kelvin waves (…"

Corrected in the text.

3. Line 121: There should be a space between "..twice a day" and "(at 0000UTC and 1200UTC)"

Corrected in the text.

4. Line 129: "…(see Figure 1a)"

Corrected in the text.

5. Line 131: "…coupled waves (Wheeler and Kiladis, 1999)"

Corrected in the text.

6. Line 135 and Figure 1: The MJO is not labeled in the Figure 1.

7. Line 167: "Fig. 1a for a …"

Corrected in the text.

8. Line 181: "…shown on Figure 2…"

Corrected in the text.

9. Line 250: "(Figures 5 and 6)…"

Corrected in the text.

10. Lines 267-271: This paragraph discusses the results from Figures 5d and 6d. Thus, I would suggest labeling 'Figure 5d' and 'Figure 6d' to make the text clearer for readers. In addition, in Line 271, I believe the authors meant to refer to MRG-TD1 rather than MRG-TD2 waves (at 3°N).

Corrected in the text.

11. Lines 276 and 431: I would suggest that ER waves are the main driver of TCWV variance near the equator and "north of 20 N", because one of their peaks is observed within 20 N-25N (Figures 5b and 6b).

Corrected in the text.

12. Line 281: According to Figure 5b, I suspect the authors meant to refer to "except for TCWV north of the AEJ above the continent in September 2021"

Corrected in the text.

13. Line 287: It is for ER event, so it should be "(see Figure 7c)" rather than Figure 7a.

Corrected in the text.

14. Figure 7 caption: I think that the shadings are TCWV "anomalies", which should be noted in caption. In Lines 3-4 of Figure 7 caption, it should be "but for MRG-TD1 and MRG-TD2 waves (black and magenta contours)"

Corrected in the text.

15. Lines 344-345: How do we know TS Larry less than 300NM south of Cape Verde? Perhaps it needs to be marked as "not shown"

Precision added in the text.

16. Line 347: "…(Figure 9a).."

Corrected in the text.

17. Line 351: "…of a Kelvin wave on…"

Corrected in the text.

18. Line 477: "…; Duvel (2021); ?…" is it missing the reference?

Corrected in the text.

19. Figures 11-13. Line labels below the panel a and b are different. (Below panel b) Dashed lines should be dew point, right?

Indeed the dashed line represented dew point. However, to simplify the discussion, reanalysis panels are deleted from figure 11-13.

Reviewer #3

Unfortunately, I found the paper to be quite a challenge to read. I think it could be improved if it was shortened with greater focus on the more important results and perhaps spreading the current content across more than one paper. I also found some of the analysis and results confusing, as detailed below. I hope the authors can consider these points to generate a new and improved manuscript for journal submission.

The paper has been extensively restructured. Especially, a better distinction was made in the text between the impacts of the tropical waves on the structure of the atmosphere and on tropical cyclogenesis. The mention to tropical cyclogenesis has been added in the tittle. Some paragraph were added to improve the clarity of the message and some figures have been removed or simplified.

**Specific comments**

Abstract, line 1 and elsewhere. In most previous on tropical waves the acronym TD has referred to Tropical Depression, not Tropical Disturbance.

Corrected in the text.

Abstract, line 3. What does it mean for the MRG-TD tracks to be "mingled in the literature"?

Clarified in the text:

> Methods of the literature struggle to distinguish the two MRG-TD tracks.

Abstract, lines 11-14. The title and the previous text of the abstract suggest that the paper is about dust and atmospheric thermodynamics only, so this inclusion of a discussion of tropical cyclogenesis seems off-topic.

Title changed to better reflect the content of the study, including tropical cyclogenesis.

Introduction, line 87. You say that you follow the method of Wheeler and Kiladis (1999) but the "protocol" of Janiga et al. (2018). Please include a sentence saying how the method of Janiga et al. differs from Wheeler and Kiladis (1999).

Corrected in the text.

Section 2, line 129. What do you mean by the "Real" shallow water model?

Replaced by "shallow water model" in the text.

Line 139. The important point to add here is that the wavenumber-frequency filtering will output wave-like structures from input that is purely red-noise, so it is not always straightforward to attribute the output to a theoretical wave.

Precision added in the text:

> Though the method can be sensitive to wave pattern generated by red noise, it allows the attribution of each component of the signal to a theoretical wave.

Figure 1a. In the introduction you said that you used the protocol of Janiga et al. (2018), but they used different regions of wavenumber and frequency for filtering as displayed in this figure. What did you use?

Precision added in the text:

> "The protocol described in Janiga et al. (2018), that consists in taking two years of data, followed by two years of zero before computing the Fourier Transform."

Figures 1b and 1c. The period 2018-2022 is clearly not long enough to get a clear picture of the waves in the spectrum. I suggest you use a longer period. Also, does your spectrum use data from all longitudes, or is it focussed on the Africa-Atlantic region?

In the revised version of the MS, the period of analysis was extended to 2012-2022.

Line 158. Say: "Figure 1 shows that the wavenumber-frequency domains of MRG and TDs do overlap significantly".

Corrected in the text.

Line 191. Here you say that you base the composite horizontal structure on the TCWV data averaged for 0-20˚N but at line 170 you said 0-15˚N. Which is correct?

The base is 0-20°N. Corrected in the text.

Figure 3. I am confused by the composites presented in Figure 3. Why aren't the TCWV structures more symmetric or anti-symmetric about the equator? If you do the wave filtering using the

symmetry constraints of Wheeler and Kiladis (1999), as implied by Figure1bc, then I expect that the TCWV composites should be more symmetric (or anti-symmetric) than what they are.

Following Janniga method, the TCWV is filtered independently for each latitude, therefore no criterion of symetry is applied.

Figure 3cd. I see nothing in this composite structure that resembles a theoretical Kelvin wave. I think you are instead looking at an artefact of the red noise background. My conclusion that it is an artefact is supported by Figure 5 that shows the Kelvin wave signal is maximized at about 30˚N, which is unlike the theoretical Kelvin wave structure which should be within about 15 degrees of the equator for the equivalent depths that you are considering.

The Kelvin composite is now computed for averaged TCWV Kelvin signal averaged between 10°S and 10°N instead of 0°-20°N before. It shows a more consistent Kelvin structure.

Figure 3 caption. You say that the number of events for the composites is included in the titles, but I don't see that.

Reference to the figure title removed in the caption. The number of event is indeed in the caption only.

Figure 5. As I said above, I do not trust the results presented for the Kelvin wave. I think it is an artefact of the method.

The filter is applied independently at each latitude, therefore the method cannot be responsible for meridional coherence of the signal. If red noise were to generate a signal, no such meridionaly coherent structure would be observed.

Changing the window on the basis of which the horizontal composite is computed allowed us to better identify a theoritical Kelvin wave. Its dynamics especially, that tends to increase the monsoon above the continent in the humid phase may explain that unexpected peak at 18°N. A smaller peak is present on all panels at 5°N, which is consistent with the findings of Tulich et al (2011). Explanation added in the MS L243-249.

Figure 5. The caption says it is the "Relative importance of tropical wave signals". Rather, it is the Squared correlation coefficient between the filtered wave signals and the associated variable. It should be noted that this is likely an overestimate of the relative importance given that there is likely a lot of noise that also contributes to the variability in the wave filtered fields.

Caption modified according to the recommendation.

Figure 6. Once again, the location of the peak of the Kelvin wave "relative importance" does not make physical sense with regards to the Kelvin wave structure which should be centred near the equator.

Same discussion as for figure 5.

Figure 7. You have labels for tropical storms Rose, Peter, and Pierre-Henri, but you do not show their actual genesis time/location or their track, just a vague timing.

Trajectories of the convective system and genesis location have been added to the Hovmöller.

Line 344. You mention TS Larry here, but it is not shown on Figure 7.

Label for Larry added in Figure 7.

Conclusions versus title. Much of the conclusions focussed on the tropical storms and TC genesis which was not reflected in the title.

Title changed to better reflect the content of the article, incl. the link with tropical cyclogenesis.

---

## Referee Report (RR1)

**Impact of Convectively Coupled TropicalWaves on the composition and vertical structure of the atmosphere above Cabo Verde in September 2021 during the CADDIWA campaign**

**Tanguy Jonville, Maurus Borne, Cyrille Flamant, Juan Cuesta, Olivier Bock, Pierre Bosser, Christophe Lavaysse, Andreas Fink, and Peter Knippertz**

This is the second round of reviews and I only have a couple very minor edits to suggest to the authors. The structure of the manuscript is much improved after the first round of revisions.

**Minor comments**

1. Page 3, line 61: "... but will remain *outside* the scope of this study...."

2. Page 4, line 106: "...aerosol optical *depth* ...)"

3. Page 5, line 130: "Information on all datasets is compiled in table 1." I would put this at the beginning of the paragraph in line 106.

**Figures**

**Figure 8:** Please specify what the dashed vs solid lines are and what red vs green lines represent in the the caption. The legend is not very clear on this (and maybe doesn't follow the notation used in the text?)

---

## Referee Report (RR2)

Although my previous concerns have been well addressed and the manuscript has been revised accordingly, I noticed several remaining issues. These do not diminish the value of this work, but I encourage the authors to address them prior to acceptance.

**Specific comments**

1. Line 63: "…in shaping extreme events Lafore et al. (2017); Peyrillé et al. (2023). → "…in shaping extreme events (Lafore et al., 2017); Peyrillé et al., 2023)."

2. Lines 74-75:  Could the authors please clarify this statement? The dependence of impact on distance remains unclear.

3. Line 91: "(2018), that…"→ "(2018) that…"

4. Line 130: "…in table 1."→ "…in Table 1."

5. Line 150: What is the "R coefficient"?

6. Lines 168-169: "…Figure 2-b and c show…" → "…Figures 2-b and c show…"

7. Lines 178-179: This statement would benefit from supporting references to demonstrate that the MJO was in an active phase during the specified three-week period.

8. Line 196: Please provide an explanation of "cycles per day (cpd)", either here or in the caption of Figure 2, to aid reader understanding.

9. Line 198: "…Figures 2b and Fig. 2c…" → "…Figures 2b and 2c…"

10. At the bottom of Figs. 1b and 1c, "WESTWARD" overlaps with "Zonal Wavenumber". Please make the necessary adjustment.

11. Line 215: "figure 3" → "Figure 3"

12. Line 243: "Figures 4b and c show" → "Figures 4c and d show"

13. Line 244: "of Matsuno (1966))" → "of Matsuno (1966)"

14. Figure 4: Please explain the unit of wind vectors.

15. Figures 4c and 4d: According to Figs. 4a, 4b, and 5, I believe the humid (dry) phase associated with the Kelvin wave should be shown in Fig. 4c (4d) rather than Fig. 4d (4c).

16. Kindly refer to the figures with more precision.

    ①  Line 287: "(Figures 6 and 7)" → "(Figures 6c and 7c)"

② Line 301: "(Figure 7)" → "(Figure 7b)"

③ Line 309: "(see Fig 6 c and d)"→ "(see Fig. 6d)"

④ Line 309: "(see Fig 7 c and d)"→ "(see Fig. 7d)"

⑤ Line 342: Which Figure result are you referring to?

17. Lines 297-299:

① A left bracket appears in Line 298, but I could not find the corresponding right bracket.

② I could not see significant southerlies over the continent in the Kelvin humid phase (Fig. 4d).

18. Line 332: "…dew-point.Schlueter…"→ "…dew-point. Schlueter…"

19. Line 340: "Figures 8d,e and e"→ "Figures 8d,e and f"

20. Figure 8: I am confused about the labels and captions. Top of Fig. 8b (8c) shows "ER- Latitude (10-15)" ("ER- Latitude (15-20)"), whereas the caption of Fig. 8 states: "(b) AEROIASI vertical dust extinction composite for the humid and dry phases of ER wave on IASI North Zone. (c) Same as (b) but for IASI South Zone." It seems the labels or captions for Figs. 8b and 8c might be swapped. In addition, there is no description of the phase represented by the line colors. This should be added for clarity.

21. Caption of Figure 9: "Number of days in IASI North Zone in dry (resp. humid) phase: 19 (resp 20). Number of days in IASI North Zone in dry (resp. humid) phase: 17 (resp 14)." It appears that the statement is repeated, or alternatively, one of them is intended to describe the South Zone. The same issue is present in the caption of Figure B1 as well.

22. Line colors in Figures 6-9 and B1: Many people are red-green colorblind, so please choose different colors.

23. The caption of Fig. 10 states that the values are averaged between 0°N and 20°N; however, Line 371 indicates "between 5°N and 20°N". Could you clarify which one is correct?

24. Lines 370-377: A statement describing the results of the Kelvin waves (Fig. 10a) appears to be missing and should be included for completeness.

25. Line 418: "…kelvin waves…"→ "…Kelvin waves…"

26. Line 508: "…TC development Hankes et al. (2015); Duvel (2021); Jonville et al. (2024a)…"→ "…TC development (Hankes et al., 2015; Duvel, 2021; Jonville et al., 2024a)…"

27. Figure 11: There is no description of the bar at the bottom of the figure; this should be added to the caption for clarity. In addition, the figure itself does not include the labels '(a)' or '(b)', even though they are referenced in the caption.

28. Figure B1: There is no description of the phase represented by the line colors. In addition, as I mentioned in my previous recommendation, the line labels below panels (a) and (b) are inconsistent. Specifically, the dashed lines below panel (b) should represent the dew point (see below).

[Figure]

---

## Author Response (AR2)

**Review of the Paper « Impact of Convectively Coupled Tropical Waves on the composition, vertical structure of the atmosphere and Tropical Cyclogenesis in the region of Cabo Verde in September 2021 during the CADDIWA campaign »**

Dear Editor

Thank you and the referees for the torough reviews.

Please find in the following the detailed answers to the reviewer's comment :

| Reviewer #1 |
| --- |

1. Page 3, line 61: "... but will remain outside the scope of this study...."

Corrected in the text.

2. Page 4, line 106: "...aerosol optical depth ...)"

Corrected in the text.

3. Page 5, line 130: "Information on all datasets is compiled in table 1." I would put this at the beginning of the paragraph in line 106.

Corrected in the text.

Figure 8: Please specify what the dashed vs solid lines are and what red vs green lines represent in the the caption. The legend is not very clear on this (and maybe doesn't follow the notation used in the text?)

Precision added in the caption. Solid line are dry temperatures, dashed lines dew point. Red lines are for humid phases, green lines (now blue to improve accessibility for coloblind people) are for dry phases.

| Reviewer #2 |
| --- |

1. Line 63: "…in shaping extreme events Lafore et al. (2017); Peyrillé et al. (2023). → "…in shaping extreme events (Lafore et al., 2017); Peyrillé et al., 2023)."

Corrected in the text.

2. Lines 74-75: Could the authors please clarify this statement? The dependence of impact on distance remains unclear.

Precision added in the text :

> The impact is also dependent on the relative location of the dust-ladden layer to the center of the convective activity: it has been found to have a catalyzing effect when being in the

> northwest quadrant further than 360km from the center of convection, but has a inhibiting effect when coming closer than 360km in the southwest quadrant.

3. Line 91: "(2018), that…"→ "(2018) that…"

Corrected in the text.

4. Line 130: "…in table 1."→ "…in Table 1."

Corrected in the text.

5. Line 150: What is the "R coefficient"?

Correlation coefficient. Corrected in the text.

6. Lines 168-169: "…Figure 2-b and c show…" → "…Figures 2-b and c show…"

Corrected in the text.

7. Lines 178-179: This statement would benefit from supporting references to demonstrate that the MJO was in an active phase during the specified three-week period.

This statement is based on the results of the wave decomposition described in the method for the MJO domain. Precision added in the text:

> result from the TCWV decomposition in the MJO domain, not shown

8. Line 196: Please provide an explanation of "cycles per day (cpd)", either here or in the caption of Figure 2, to aid reader understanding.

Added in the text.

9. Line 198: "…Figures 2b and Fig. 2c…" → "…Figures 2b and 2c…"

Corrected in the text.

10. At the bottom of Figs. 1b and 1c, "WESTWARD" overlaps with "Zonal Wavenumber". Please make the necessary adjustment.

Figure corrected.

11. Line 215: "figure 3" → "Figure 3"

Corrected in the text.

12. Line 243: "Figures 4b and c show" → "Figures 4c and d show"

Corrected in the text.

13. Line 244: "of Matsuno (1966))" → "of Matsuno (1966)"

Corrected in the text.

14. Figure 4: Please explain the unit of wind vectors.

Precision added in the figure.

15. Figures 4c and 4d: According to Figs. 4a, 4b, and 5, I believe the humid (dry) phase associated with the Kelvin wave should be shown in Fig. 4c (4d) rather than Fig. 4d (4c).

Figure corrected.

16. Kindly refer to the figures with more precision.

① Line 287: "(Figures 6 and 7)" → "(Figures 6c and 7c)"

② Line 301: "(Figure 7)" → "(Figure 7b)"

③ Line 309: "(see Fig 6 c and d)" → "(see Fig. 6d)"

④ Line 309: "(see Fig 7 c and d)" → "(see Fig. 7d)"

⑤ Line 342: Which Figure result are you referring to?

Corrected in the text.

17. Lines 297-299:

① A left bracket appears in Line 298, but I could not find the corresponding right bracket.

② I could not see significant southerlies over the continent in the Kelvin humid phase (Fig. 4d).

Corrected in the text for the bracket. As for the southerlies, they are actually shown on the « dry phase » panel, as the humid phase is above the continent when the dry phase is above cabo verde. Sentence changed to improve clarity:

> see Figure 4d that shows southerlies when the Kelvin humid phase is located above the continent, increasing the monsoon flow

18. Line 332: "…dew-point.Schlueter…" → "…dew-point. Schlueter…"

Corrected in the text.

19. Line 340: "Figures 8d,e and e" → "Figures 8d,e and f"

Corrected in the text.

20. Figure 8: I am confused about the labels and captions. Top of Fig. 8b (8c) shows "ER- Latitude (10-15)" ("ER- Latitude (15-20)"), whereas the caption of Fig. 8 states: "(b) AEROIASI vertical dust extinction composite for the humid and dry phases of ER wave on IASI North Zone. (c) Same as (b) but for IASI South Zone." It seems the labels or captions for Figs. 8b and 8c might be swapped. In addition, there is no description of the phase represented by the line colors. This should be added for clarity.

Corrected in the caption. The title of the figure was right.

21. Caption of Figure 9: "Number of days in IASI North Zone in dry (resp. humid) phase: 19 (resp 20). Number of days in IASI North Zone in dry (resp. humid) phase: 17 (resp 14)." It appears that the statement is repeated, or alternatively, one of them is intended to describe the South Zone. The same issue is present in the caption of Figure B1 as well.

Corrected in the text.

22. Line colors in Figures 6-9 and B1: Many people are red-green colorblind, so please choose different colors.

Color palette changed.

23. The caption of Fig. 10 states that the values are averaged between 0°N and 20°N; however, Line 371 indicates "between 5°N and 20°N". Could you clarify which one is correct?

Corrected in the text. The caption was the right one.

24. Lines 370-377: A statement describing the results of the Kelvin waves (Fig. 10a) appears to be missing and should be included for completeness.

Sentence on Kelvin wave added in the text:

25. Line 418: "…kelvin waves…"→ "…Kelvin waves…"

Corrected in the text.

26. Line 508: "…TC development Hankes et al. (2015); Duvel (2021); Jonville et al. (2024a)…"→ "…TC development (Hankes et al., 2015; Duvel, 2021; Jonville et al., 2024a)…"

Corrected in the text.

27. Figure 11: There is no description of the bar at the bottom of the figure; this should be added to the caption for clarity. In addition, the figure itself does not include the labels '(a)' or '(b)', even though they are referenced in the caption.

Description of the bars added in the caption.

28. Figure B1: There is no description of the phase represented by the line colors. In addition, as I mentioned in my previous recommendation, the line labels below panels (a) and (b) are inconsistent. Specifically, the dashed lines below panel (b) should represent the dew point (see below).

Corrected in the figure and caption.